

**1  GPU-HADVPPM V1.0: high-efficient parallel GPU design of the**

**2  Piecewise Parabolic Method (PPM) for horizontal advection in**

**3  air quality model (CAMx V6.10)**

**Kai Cao[1], Qizhong Wu[1], Lingling Wang[2], Nan Wang[2], Huaqiong Cheng[1], Xiao**
**Tang[3], Dongqing Li[1], and Lanning Wang[1]**
[1]College of Global Change and Earth System Science, Beijing Normal University,
Beijing 100875, China
[2]Henan Ecological Environmental Monitoring Centre and Safety Center, Henan Key
Laboratory of Environmental Monitoring Technology, Zhengzhou 450008, China
[3]State Key Laboratory of Atmospheric Boundary Layer Physics and Atmospheric
Chemistry, Institute of Atmospheric Physics, Chinese Academy of Science, Beijing
100029, China
**Correspondence to:** Qizhong Wu (wqizhong@bnu.edu.cn); Lingling
Wang(928216422@qq.com); Lanning Wang (wangln@bnu.edu.cn)
**Abstract.** With semiconductor technology gradually approaching its physical and thermal limits,
Graphics processing unit (GPU) is becoming an attractive solution in many scientific applications
due to their high performance. This paper presents an application of GPU accelerators in air quality
model. We endeavor to demonstrate an approach that runs a PPM solver of horizontal advection
(HADVPPM) for air quality model CAMx on GPU clusters. Specifically, we first convert the
HADVPPM to a new Compute Unified Device Architecture C (CUDA C) code to make it
computable on the GPU (GPU-HADVPPM). Then, a series of optimization measures are taken,
including reducing the CPU-GPU communication frequency, increasing the size of data
computation on GPU, optimizing the GPU memory access, and using thread and block indices in
order to improve the overall computing performance of CAMx model coupled with GPU-
HADVPPM (named as CAMx-CUDA model). Finally, a heterogeneous, hybrid programming
paradigm is presented and utilized with the GPU-HADVPPM on GPU clusters with Massage
Passing Interface (MPI) and CUDA. Offline experiment results show that running GPU-
HADVPPM on one NVIDIA Tesla K40m and NVIDIA Tesla V100 GPU can achieve up to 845.4x
and 1113.6x acceleration. By implementing a series of optimization schemes, the CAMx-CUDA
model resulted in a 29.0x and 128.4x improvement in computational efficiency using a GPU



accelerator card on a K40m and V100 cluster, respectively. In terms of the single-module
computational efficiency of GPU-HADVPPM, it can achieve 1.3x and 19.4x speedup on NVIDIA
Tesla K40m GPU and NVIDA Tesla V100 GPU respectively. The multi-GPU acceleration algorithm
enables 4.5x speedup with 8 CPU cores and 8 GPU accelerators on V100 cluster.
**1.   Introduction**

Since the introduction of the personal computer in the late 1980s, the computer

and mobile device industry has been one of the most flourishing markets all over the
world (Bleichrodt et al., 2012). In recent years, the improvement of the performance of
the Central Processing Unit (CPU) is limited by its heat dissipation, the development
of Moore's Law has flattened. A common trend in high-performance computing today
is the utilization of hardware accelerators that execute codes rich in data parallelism to
form high-performance heterogeneous system. GPUs are widely used as accelerators
due to high peak performance offered. In the top ten supercomputing list released in
December 2022 (https://www.top500.org/lists/top500/list/2022/11/, last access: 19
December 2022), there are seven heterogeneous supercomputing platforms built with
CPU processors and GPU accelerators, of which the top one Frontier at the Oak Ridge
National Laboratory uses AMD's third-generation EPYC CPU and AMD Instinct
MI250X GPU, and its computing performance reaches Exascale ($10^{18}$ calculations per
second) for the first time (https://www.amd.com/en/press-releases/2022-05-30-world-
s-first-exascale-supercomputer-powered-amd-epyc-processors-and-amd, last access:
19 December 2022). Such powerful computing performance of the heterogeneous
system not only injects new vitality into high-performance computing, but also provides
new solutions for improving the performance of numerical models in geoscience.

The GPU has proven successful in weather models such as Non-Hydrostatic

Icosahedral Model (NIM; Govett et al.,2017), Global/Regional Assimilation and
Prediction System (GRAPES; Xiao et al., 2022), and Weather Research and Forecasting
model (WRF; Huang et al., 2011; Huang et al., 2012; Mielikainen et al., 2012a;



Mielikainen et al., 2012b; Mielikainen et al., 2013a ; Mielikainen et al., 2013b; Price et
al., 2014; Huang et al., 2015), ocean models such as LASG/IAP Climate System Ocean
Model (LICOM; Jiang et al., 2019; Wang et al., 2021a) and Princeton Ocean Model
(POM; Xu et al., 2015), and the Earth System Model of Chinese Academy of Sciences
(CAS-EMS; Wang et al., 2021b ; Wang et al., 2021c).
Govett et al., (2017) used Open Accelerator (OpenACC) directives to port the
dynamics of NIM to the GPU and achieved 2.5x acceleration. Also using OpenACC
directives, Xiao et al., (2022) ported the PRM (Piecewise Rational Method) scalar
advection scheme in the GRAPES to the GPU, achieving up to 3.51x faster than 32
CPU cores. In terms of the most widely used WRF, several parameterization schemes,
such as RRTMG_LW scheme (Price et al., 2014), 5-layer thermal diffusion scheme
(Huang et al., 2015), Eta Ferrier Cloud Microphysics scheme (Huang et al., 2012),
Goddard Shortwave scheme (Mielikainen et al., 2012a), Kessler cloud microphysics
scheme (Mielikainen et al., 2013b), SBU-YLIN scheme (Mielikainen et al., 2012b),
WMS5 scheme (Huang et al., 2011), WMS6 scheme (Mielikainen et al., 2013a), etc.,
have been ported heterogeneously using CUDA C and achieved 37x~896x acceleration
results. The LICOM has carried out heterogeneous porting using OpenACC (Jiang et
al., 2019) and Heterogeneous-compute Interface for Portability C (HIP C) technologies,
and achieved up to 6.6x and 42x acceleration, respectively (Wang et al., 2021a). For the
Princeton Ocean Model, Xu et al., (2015) use CUDA C to carry out heterogeneous
porting and optimization, the performance of gpu-POM v1.0 on four GPUs is
comparable to that on 408 standard Intel Xeon X5670 CPU cores. In terms of climate
system model, Wang et al., (2021c) and Wang et al., (2021b) used CUDA Fortran and
CUDA C to carry out heterogeneous porting of the RRTMG_SW and RRTMG_LW
scheme of the atmospheric component model of the CAS-EMS earth system model,
and achieved a 38.88x and 77.78x acceleration respectively.
Programming a GPU accelerator can be a hard and error-prone process that
requires specially designed programing methods, there are three widely used methods
for porting program to GPUs as described above. The first method uses the OpenACC



directive (https://www.openacc.org/, last access: 19 December 2022) which provides a
set of high-level directives that enable C/C++ and Fortran programmers to utilize
accelerators. The second method uses CUDA Fortran. CUDA Fortran is a software
compiler which co-developed by the Portland Group (PGI) and NVIDIA, and tool chain
for building performance optimized GPU-accelerated Fortran applications targeting the
NVIDIA GPU platform (https://developer.nvidia.com/cuda-fortran, last access: 19
December 2022).  CUDA C involves rewriting the entire program using standard C
programming      language      and      low-level      CUDA      subroutines
(https://developer.nvidia.com/cuda-toolkit, last access: 19 December 2022) to support
the NVIDIA GPU accelerator. Compared to the other two technologies, CUDA C
porting scheme is the most complex, but its computational performance is the highest
(Mielikainen et al., 2012b; Wahib and Maruyama, 2013; Xu et al., 2015).
Air quality models are critical to understanding how the chemistry and
composition of atmospheric may change over 21$^{st}$ century, as well as preparing adaptive
responses or developing mitigation strategies. Because air quality models need to take
into account the complex physicochemical processes that occur in the atmosphere of
anthropogenic and naturally emissions, simulations are computationally expensive.
Compared to the other geoscientific numerical models, few research have carried out
heterogeneous porting of air quality models. In this study, CUDA C scheme was
implemented in this paper to carry out the hotspot module porting attempt of CAMx in
order to improve the computation efficiency.

## 2.   The CAMx model and experiments

### 2.1.   Model description

CAMx model is a state-of-the air quality model developed by Ramboll Environ
(https://www.camx.com/, last access: 19 December 2022). CAMx version 6.10 (CAMx
V6.10; ENVIRON, 2014) is chosen in this study, it simulates the emission, dispersion,
chemical reaction, and removal of pollutants by marching the Eulerian continuity



equation forward in time for each chemical species on a system of nested three-
dimensional grids. The Eulerian continuity equation is expressed mathematically in
terrain-following height coordinates as formula (1):
$$\frac{\partial c_i}{\partial t} = -\nabla_H \cdot V_H c_i + \left[ \frac{\partial(c_i \eta)}{\partial z} - c_i \frac{\partial^2 h}{\partial z \partial t} \right] + \nabla \cdot \rho K \nabla(c_i/\rho)$$

$$+ \left. \frac{\partial c_i}{\partial t} \right|_{Emission} + \left. \frac{\partial c_i}{\partial t} \right|_{Chemistry} + \left. \frac{\partial c_i}{\partial t} \right|_{Removal} \tag{1}$$

$$\nabla_H \cdot \rho V_H = \frac{m^2}{A_{yz}} \frac{\partial}{\partial x} \left( \frac{u A_{yz} \rho}{m} \right) + \frac{m^2}{A_{xz}} \frac{\partial}{\partial y} \left( \frac{v A_{xz} \rho}{m} \right) \tag{2}$$

The first term on the right-hand side represents horizontal advection. In the
numerical methods, the equation of horizontal advection (described in formula (2)) is
performed using the area preserving flux-form advection solver of the Piecewise
Parabolic Method (PPM) of Colella and Woodward (1984) as implemented by Odman
and Ingram (1996).  The PPM solution of horizontal advection (HADVPPM) was
incorporated into CAMx model because it provides higher order accuracy with minimal
numerical diffusion.
In the Fortran code implementation of HADVPPM scheme, the CAMx main
program calls the emistrns program, which mainly performs the physical processes such
as emission, diffusion, advection and dry/wet deposition of pollutants. And then, the
horizontal advection program is invoked by emistrns program to solve the horizontal
advection equation by using the HADVPPM scheme.
**2.2.  Benchmark performance experiments**
The first step of the porting is to test the performance of CAMx benchmark version
and identify the hotspots of the model. On the Intel x86 CPU platform, we launch two
processes concurrently to run the CAMx and take advantage of the Intel Trace Analyzer
Collector(ITAC;        https://www.intel.com/content/www/us/en/docs/trace-analyzer-
collector/get-started-guide/2021-4/overview.html, last access: 19 December 2022) and
Intel                                                                        VTune



Profiler(VTune;https://www.intel.com/content/www/us/en/develop/documentation/vtu
ne-help/top.html, last access: 19 December 2022) performance analysis tools to collect
performance information during CAMx operation.
The general MPI performance can be reported by the ITAC tool, and MPI load
balance information, computation and communication profiling of each process is
shown as Fig. 1a. During the running process of CAMx model, Process 0 (P0) spends
99.6% of the time on the MPI_Barrier function and only 0.4% of the time on
computation, while Process 1(P1) spends 99.8% of its time computation and only 0.2%
of its time receiving messages from P0. It is indicated that the parallel design of CAMx
model adopts Master-Slave mode, P0 is responsible for inputting and outputting data
and calling the MPI_Barrier function to synchronize the process, so there is a lot of
MPI waiting time. The other processes are responsible for computation.
The VTune tool is used to detect the runtime of each module and the most time-
consuming functions on P1. As shown in Figure 1b, the top four time-consuming
modules are chemistry, diffusion, horizontal advection, and vertical advection in CAMx
model. The top five most time-consuming programs and their elapsed time are in Table
1. The total runtime of P1 is 325.1 seconds, and the top five most time-consuming
programs are ebirate, hadvppm, tridiag, diffus, and ebisolv program. Top1 and Top2's
most time-consuming programs take 49.4 and 35.6 seconds, respectively. By viewing
the Fortran code of the above programs, the hadvppm program has few calculation
branches, and its calculation process does not involve iterative operations, which
satisfies the basic conditions for the program to run on the GPU. Therefore, a GPU
acceleration version of the HADVPPM scheme, namely GPU-HADVPPM, is built to
improve CAMx performance.



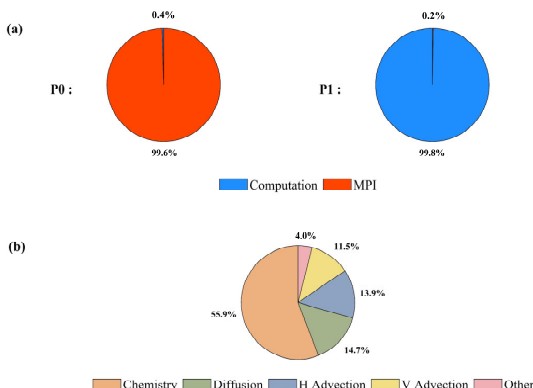


**Figure 1.** The computation performance of the modules in the CAMx model. (a) Computation and

communication profiling of P0 and P1. (b) Overhead proportions of P1. The top four most time-

consuming modules are chemistry, diffusion, horizontal advection, and vertical advection.

**Table 1.** The top five most time-consuming programs on the P1 (Total runtime is 325.1 seconds).

|  | Program | Module | CPU time (s) |
|---|---|---|---|
| **Top1** | *ebirate.f* | Chemistry | 49.4 |
| **Top2** | *hadvppm.f* | Horizontal advection | 35.6 |
| **Top3** | *tridiag.f* | Vertical advection, Diffusion | 28.0 |
| **Top4** | *diffus.f* | Diffusion | 26.9 |
| **Top5** | *ebisolv.f* | Chemistry | 26.2 |

## 2.3. Porting scheme introduction

The heterogeneous scheme of CAMx-CUDA is shown in Figure 2. The second

time-consuming program hadvppm in CAMx model, was selected to implement the

heterogeneous porting. In order to map the hadvppm program to the GPU, the Fortran

code of hadvppm program is converted to standard C code. Then, CUDA programing

language which is tailor-made for NVIDIA was added to convert the standard C code



into CUDA C for data-parallel execution on GPU, as GPU-HADVPPM. It prepares the
input data for GPU-HADVPPM by constructing random numbers, and tests its offline
performance on GPU platform.

After coupling GPU-HADVPPM to CAMx model, the advection module code was

optimized according to the characteristics of GPU architecture to improve the overall
computational efficiency on CPU-GPU heterogeneous platform. And then, the multi-
CPU core and multi-GPU card acceleration algorithm was adopted to improve the
parallel extensibility of heterogeneous computing. Finally, the coupling performance
test is implemented after verifying the different CAMx model simulation results.

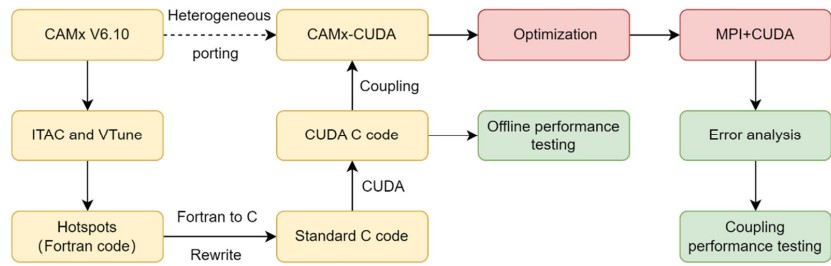


**Figure 2.** Heterogeneous porting scheme of CAMx-CUDA model.
**2.4.   Hardware components and software environment of the testing system**

The experiments are conducted on two GPU clusters: K40m and V100.

hardware components and software environment of the two clusters are listed in Table
2. The K40m cluster is equipped with two 2.5GHz 16-core Intel Xeon E5-2682 v4 CPU
processors and one NVIDIA Tesla K40m GPU card on each node. The NVIDIA Tesla
K40m GPU has 2880 CUDA cores with 12GB of memory. The V100 cluster contains
two 2.7GHz 24-core Intel Xeon Platinum 8168 processors and eight NVIDIA Tesla
V100 GPU cards with 5120 CUDA cores and 16GB memory on each card.
**Table 2.** Configurations of GPU cluster.

| Hardware components | |
|---|---|
| **CPU** | **GPU** |





| K40m cluster | Intel Xeon E5-2682 v4 CPU @2.5GHz, 16 cores | NVIDIA Tesla K40m, 2880 CUDA cores, 12GB memory |
|---|---|---|
| V100 cluster | Intel Xeon Platinum 8168 CPU @2.7 GHz, 24 cores | NVIDIA Tesla V100, 5120 CUDA cores, 16GB memory |
| **Software environment** | | |
| | **Compiler and MPI** | **Programming Model** |
| **K40m cluster** | Intel-2021.4.0 | CUDA-10.2 |
| **V100 cluster** | Intel-2019.1.144 | CUDA-10.0 |

For Fortran and standard C programming, Intel Toolkit (including compiler and
MPI library) version 2021.4.0 and version 2019.1.144 are employed for compiling on
Intel Xeon E4-2682 v4 CPU and Intel Xeon Platinum 8168 CPU, respectively. And
then, CUDA version 10.2 and version 10.0 are employed on NVIDIA Tesla K40m GPU
and NVIDIA Tesla V100 GPU. CUDA (NVIDIA, 2020) is an extension of the C
programming language that offers direct programming of the GPUs. In CUDA
programming, what is called a kernel is actually a subroutine that can be executed on
the GPU. The underlying code in the kernel is divided into a series of threads, each with
a unique "ID" number that can simultaneously process different data through a single-
instruction multiple-thread (SIMT) parallel mode. These threads are grouped into
equal-sized thread blocks, which are organized into a grid.
**3.   Porting and optimization of CAMx advection module on heterogeneous**
**platform**
**3.1.   Mapping HADVPPM scheme to GPU**
**3.1.1.   Manual code translation from Fortran to standard C**
As the CAMx V6.10 code was written in Fortran 90, we rewrote the hadvppm
program from Fortran to CUDA C. As an intermediate conversion step, we refactor the
original Fortran code using standard C. During the refactoring, some considerations are
listed in Table 3:
(1) The subroutine name refactored with standard C must be followed by an





underscore identifier, which can only be recognized when Fortran calls.

(2) In Fortran language, the parameters are transferred by memory address by

default. In the case of mixed programming in Fortran and standard C, parameters
transferred by Fortran are processed by the pointer in standard C.

(3) Variable precision types defined in standard C must be strictly consistent with

those in Fortran.

(4) Some built-in functions in Fortran are not available in standard C and need to

be defined in standard C macro definitions.

(5) For multidimensional arrays, Fortran and standard C follow column-major and

row-major order in-memory read and write, respectively;

(6) Array subscripts in Fortran and standard C are indexed from any integer and 0,

respectively.
**Table 3.** Some considerations during Fortran to C refactoring.

| | **Fortran code** | **C code** |
|---|---|---|
| **Function name** | *subroutine hadvppm()* | *void hadvppm()* |
| **Parameter passing** | *hadvppm(nn,dt,dx,con,vel,area,areav, flxarr,mynn)* | *hadvppm(int \*nn,float \*dt, float \*dx, float \*con, float \*vel, float \*area, float \*areav, float \*flxarr, int \*mynn)* |
| **Variable precision** | *real(kind=8) x* | *double x* |
| **Built-in functions** | *max* | *#define Max(a, b) ((a)>(b)?(a):(b))* |
| **Memory read and write for multidimensional array** | Column-major | Row-major |
| **Array subscript index** | Starting from any integer | Starting from 0 |


### 3.1.2.  Converting standard C code into CUDA C

After refactoring the Fortran code of the hadvppm program with standard C,



CUDA was used to convert the C code into CUDA C to make it computable on the
GPU. A standard C program using CUDA extensions distributes a large number of
copies of the kernel functions into available multiprocessors and executes them
simultaneously on the GPU.

Figure 3 shows the implementation process of the GPU-HADVPPM. As
mentioned in Sect.2.1, xyadvec program calls the hadvppm program to solve the
horizontal advection function. Since the rewritten CUDA program cannot be called
directly by Fortran program (xyadvec.f), we add an intermediate subroutine
(hadvppm.c) as an interface to transfer the parameters and data required for GPU
computing from xyadvec Fortran program to hadvppm_kernel CUDA C program.

A CUDA program automatically uses numerous threads on GPU to execute kernel
functions. Therefore, the hadvppm_kernel CUDA C program first calculates the
number of parallel threads according to the array dimension. And then allocate GPU
memory, and copy parameters and data from the CPU to the GPU. As the CUDA
program launches a large number of parallel threads to execute kernel functions
simultaneously, the computation results will be copied from the GPU back to the CPU.
Finally, the GPU memory is released, and data computed on the GPU is returned to the
xyadvec program via hadvppm C program.

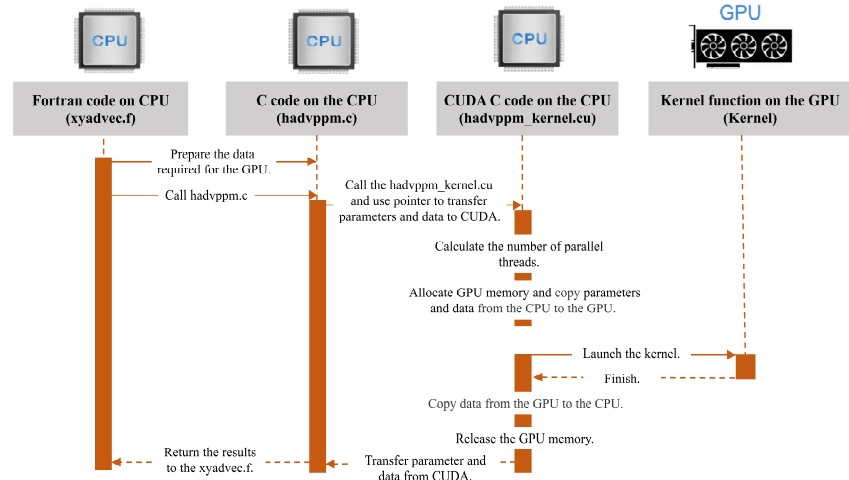


**Figure 3.** The calling and computation process of the GPU-HADVPPM on the CPU-GPU



heterogeneous platform.

### 3.2.    Coupling and optimization of GPU-HADVPPM scheme on a single GPU

After the hadvppm program was rewritten with standard C and CUDA, the
implementation process of HADVPPM scheme is loaded from the CPU to the GPU.
And then, we coupled the GPU-HADVPPM to CAMx model. For ease of description,
we will refer to this original heterogeneous version of CAMx as CAMx-CUDA V1.0.
In the CAMx-CUDA V1.0, four external loops are nested when hadvppm C program is
called by the xyadvec program. It will result in the widespread data transfers from the
CPU to the GPU over the PCIe bus within a time step, making the computation of the
CAMx-CUDA V1.0 inefficient.
Therefore, we optimize the xyadvec Fortran program to significantly reduce the
frequency of data transmission between CPU and GPU, increase the amount of data
computation on GPU, and improve the total computing efficiency of the CAMx on
CPU-GPU heterogeneous platforms. In the original CAMx-CUDA V1.0, four external
loops outside of hadvppm C program and several one-dimensional arrays are computed
before calling hadvppm C program. Then the CPU will frequently launch the GPU and
transfer data to it within a time step. When the code optimization is completed, three or
four-dimensional arrays required for GPU computation within a time step will be sorted
before calling the hadvppm C program, and then the CPU will package and transfer the
arrays to the GPU in batches. The example of xyadvec Fortran program optimization
was shown in Figure S1.
The details of four different versions are shown in Table 4. In the CAMx-CUDA
V1.0, the Fortran code of the HADVPPM scheme was rewritten using standard C and
CUDA, and the xyadvec program is not optimized. The dimensions of the c1d variable
array transmitted to GPU in the X and Y directions are 157 and 145 in this case,
respectively. In CAMx-CUDA V1.1 and CAMx-CUDA V1.2, the c1d variable
transmitted from CPU to GPU are expanded to two (about 23,000 numbers) and four
dimensions (about 27.4 million numbers) by optimizing the xyadvec Fortran program





and hadvppm_kernel CUDA C program, respectively.

The order in which data is accessed in GPU memory affects the computational

efficiency of the code. In the CAMx-CUDA V1.3 of the Table 4, we further optimized
the order in which data is accessed in GPU memory based on the order in which it is
stored in memory, and eliminated unnecessary assignment loops that were added due
to the difference in memory read order between Fortran and C.

As described in Sect.2.4, a thread is the basic unit of parallelism in CUDA

programming. The structure of threads is organized into a three-level hierarchy. The
highest level is a grid, which consists of three-dimensional thread blocks. The second
level is a block, which also consists of three-dimensional threads. Built-in CUDA
variable *threadIdx.x* determines a unique thread "ID" number inside a thread block.
Similarity, built-in variable *blockIdx.x* and *blockIdx.y* determine which block to execute
on, and the size of the block is determined by using the built-in variable *blockdim.x*.
For the two-dimensional horizontal grid points, many threads and blocks can be
organized so that each CUDA thread computes the results for different spatial positions
simultaneously.

Before the CAMx-CUDA V1.4, the loops for three-dimension spatial grid points

(i,j,k) are replaced by index computations only using thread index (*i = threadIdx.x +*
*blockIdx.x\*blockDim.x*), to use thread indexes only computes the grid point in the x or
y direction simultaneous. In order to take full advantage of thousands of threads in the
GPU, we implement thread and block indices (*i = threadIdx.x + blockIdx.x\*blockDim.x;*
*j = blockIdx.y*) to compute all horizontal grid points (*i,j*) simultaneous in the CAMx-
CUDA V1.4. This is permitted because there are no interactions among horizontal grid
points.
**Table 4.** The details of different CAMx-CUDA versions during optimization.

| Version | Major revisions | Amount of data computation on GPU |
| --- | --- | --- |
| **CAMx-CUDA V1.0** | The Fortran code of the HADVPPM subroutine was rewritten using standard C and CUDA, and *xyadvec.f* was not optimized. | 157 and 145 in the x direction and y direction for the c1d variable, respectively. |
| **CAMx-CUDA V1.1** | Optimize *xyadec.f* and | 157×145, |





| | | |
|---|---|---|
| | *hadvppm_kernel.cu* to expand the dimension of the array transmitted to the GPU from 1-dimensional to 2-dimensional. | about 23,000 numbers for the c2d variable. |
| **CAMx-CUDA V1.2** | Based on the CAMx-CUDA V1.1, the dimension of the array transmitted to the GPU is extended from 2 to 4 dimensions. | 157×145×14×86, about 27.4 million numbers for the c4d variable. |
| **CAMx-CUDA V1.3** | Based on the CAMx-CUDA V1.2, the order of GPU memory access is optimized and unnecessary assignment loops are eliminated. | 157×145×14×86, about 27.4 million numbers for the c4d variable. |
| **CAMx-CUDA V1.4** | Based on the CAMx-CUDA V1.3, using thread and block indices ($i = threadIdx.x + blockIdx.x*blockDim.x; j = blockIdx.y$). | 157×145×14×86, about 27.4 million numbers for the c4d variable. |


### 3.3.  MPI+CUDA acceleration algorithm of CAMx-CUDA on multiple GPUs


Generally, super-large clusters have thousands of compute nodes. The current CAMx V6.10, implemented by adopting MPI communication technology, typically runs on dozens of compute nodes. Once the GPU-HADVPPM is coupled into the CAMx, it also has to run on multiple compute nodes which equipped one or more GPUs on each node. To make full use of multi-core and multi-GPU supercomputers and further improve the overall computational performance of the CAMx-CUDA, we adopt a parallel architecture with an MPI+CUDA hybrid paradigm, that is, the collaborative computing strategy of multiple CPU cores and multiple GPU cards is adopted during the operation of CAMx-CUDA model. Adopt this strategy, the GPU-HADVPPM can run on multiple GPUs, the Fortran code of other modules in CAMx-CUDA model can run on multiple CPU cores.

As is shown in Figure 4., after the simulated region is subdivided by MPI, a CPU core is responsible for the computation of a subregion. In order to improve the total computational performance of the CAMx-CUDA model, we further used the NVIDIA CUDA library to obtain the number of GPUs per node, and then used MPI process ID and remainder function to determine the GPU ID to be launched by each node. Finally,



we used NVIDIA CUDA library cudaSetDevice to configure a GPU card for each CPU
core.

According to the benchmark performance experiments, the parallel design of

CAMx adopts Master-Slave mode, P0 is responsible for inputting and outputting data.
If two processes (P0 and P1) were launched, only the P1 and its configured GPU
participate in integration.

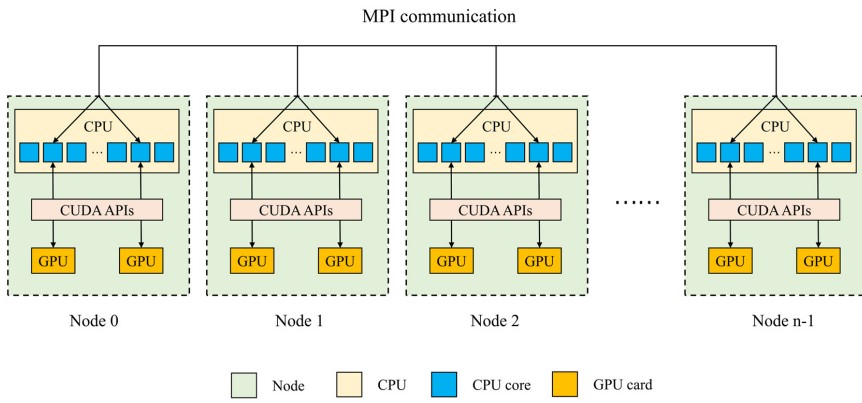


**Figure 4.** An example of parallel architecture with an MPI+CUDA hybrid paradigm on multiple
GPUs.
**4.    Experimental results**

The validation and evaluation of porting the HADVPPM scheme from CPU to

GPU platform were conducted using offline and coupling performance experiments.
First, we validate the result between different CAMx versions, and then the offline
performance of the GPU-HADVPPM on a single GPU was tested by offline experiment.
Finally, the coupling performance experiments illustrate its potential in three
dimensions with varying chemical regimes. In Sect.4.2 and Sect.4.4, the CAMx version
of the HADVPPM scheme written by Fortran language, standard C, and CUDA C are
named as F, C, and CUDA C, respectively.



### 4.1. Experimental setup


344  The test case is a 48h simulation covering the Beijing, Tianjin and part region of

345 Hebei province. The horizontal resolution is 3km with $145 \times 157$ grid boxes. The

346 model adopted 14 vertical layers. The simulation started at 12:00 UTC, 01 November

347 2020, and ended at 12:00 UTC, 03 November 2020. The meteorological fields driving

348 the CAMx model were provided by the Weather Research and Forecasting (WRF;

349 Skamarock et al., 2008) model. The Sparse Matrix Operator Kernel Emission (SMOKE;

350 Houyoux and Vukovich, 1999) version 2.4 model is used to provide gridded emission

351 data for the CAMx model. The emission inventories (Sun et al., 2022) include the

352 regional emissions in East Asia that were obtained from the Transport and Chemical

353 Evolution over the Pacific (TRACE-P; Streets et al., 2003; Streets et al., 2006) project,

354 30-min spatial resolution Intercontinental Chemical Transport Experiment-Phase B

355 (INTEX-B; Zhang et al., 2009) and the updated regional emission inventories in North

356 China. The physical and chemical numerical methods selected during CAMx model

357 integration are listed in Table S2.

### 4.2. Error analysis

359  The hourly concentration of different CAMx simulations (Fortran, C, and CUDA

360 C versions) are compared to verify the usefulness of the CUDA C version of CAMx for

361 the numerical precision of scientific usage. Here, we chose six major species, i.e., $SO_2$,

362 $O_3$, $NO_2$, CO, $H_2O_2$ and $PSO_4$ after 48h integration to verify the results.

363  Due to the differences in programming languages and hardware, the simulation

364 results are affected during the porting process. Figure 5~7 present the spatial

365 distribution of $SO_2$, $O_3$, $NO_2$, CO, $H_2O_2$ and $PSO_4$, as well as the absolute errors (AEs)

366 of their concentrations from different CAMx versions. The species' spatial patterns of

367 the three CAMx versions are visually very similar. Especially between the Fortran and

368 C versions, the AEs in all grid boxes are in the range of ±0.01 ppbV (the unit of $PSO_4$

369 is $\mu g \cdot m^{-3}$). During the porting process, the primary error comes from converting



standard C to CUDA C. In general, for $SO_2$, $O_3$, $NO_2$, $H_2O_2$ and $PSO_4$, the AEs in the
majority of grid boxes are in the range of $\pm 0.8$ ppbV or $\mu g \cdot m^{-3}$ between the
standard C and CUDA C versions; for CO, because its background concentration is
higher, the AEs of standard C and CUDA C versions are outside that range which falls
into the range of -8 and 8 ppbV in some grid boxes and shows more obvious AEs than
other species.

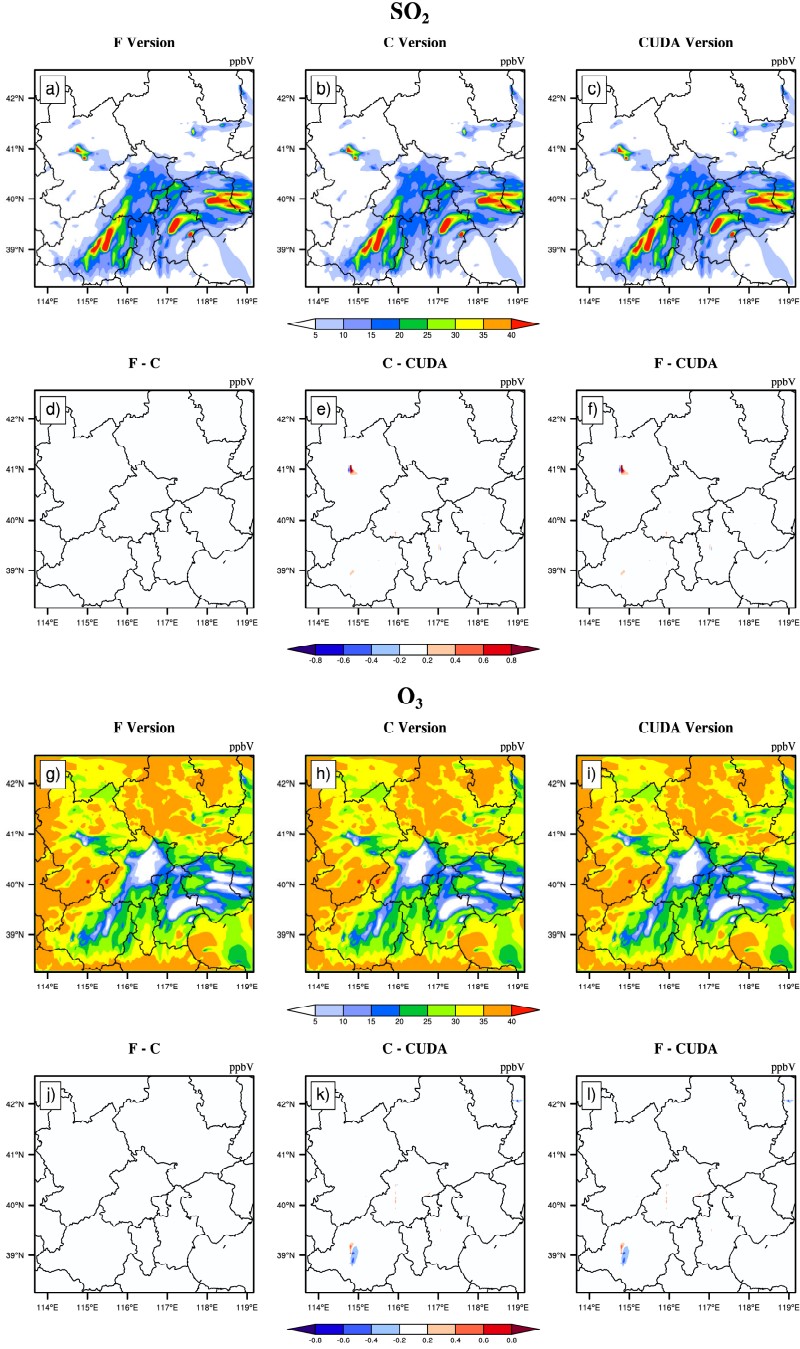

**Figure 5.** SO$_2$ and O$_3$ concentrations outputted by CAMx model for Fortran, standard C, and CUDA



C versions. Panels (a) and (g) are from Fortran versions. Panels (b) and (h) are from standard C
versions. Panels (c) and (i) are from CUDA C versions. Panels (d) and (j) are the output
concentration differences of Fortran and standard C versions. Panels (e) and (k) are the output
concentration differences of standard C and CUDA C versions. Panels (f) and (l) are the output
concentration differences of Fortran and CUDA C versions.

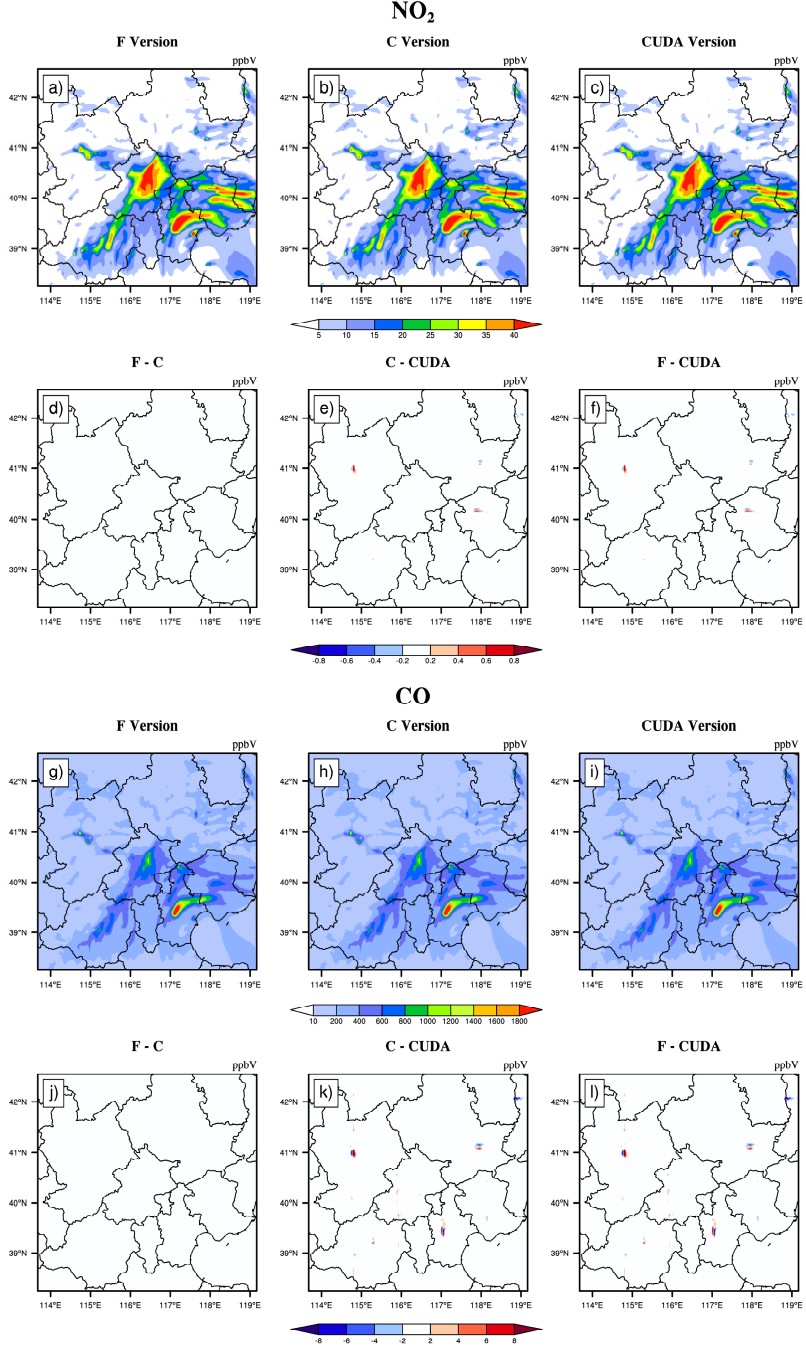


**Figure 6.** NO$_2$ and CO concentrations outputted by CAMx model for Fortran, standard C, and



CUDA C versions. Panels (a) and (g) are from Fortran versions. Panels (b) and (h) are from standard
C versions. Panels (c) and (i) are from CUDA C versions. Panels (d) and (j) are the output
concentration differences of Fortran and standard C versions. Panels (e) and (k) are the output
concentration differences of standard C and CUDA C versions. Panels (f) and (l) are the output
concentration differences of Fortran and CUDA C versions.





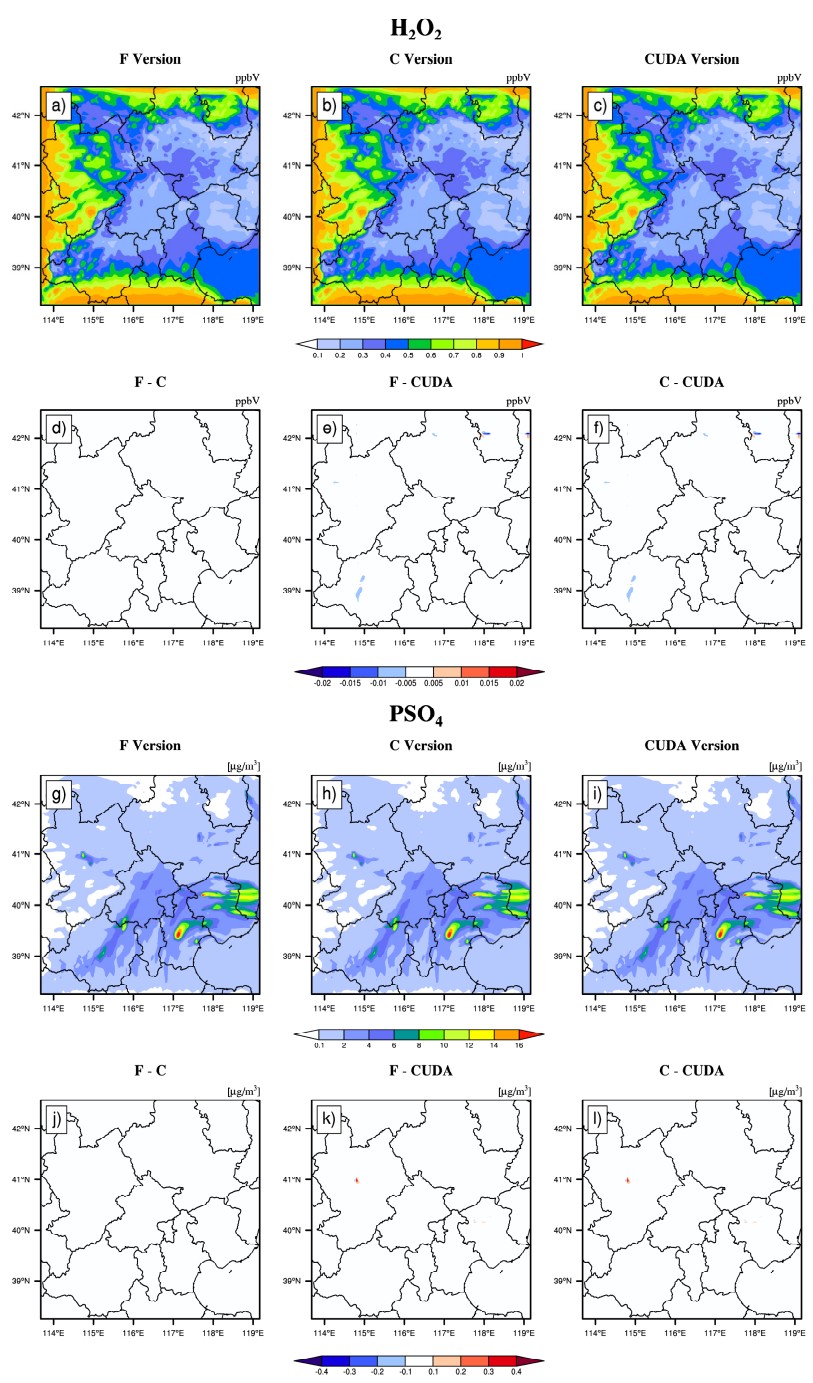




**Figure 7.** $H_2O_2$ and $PSO_4$ concentrations outputted by CAMx model for Fortran, standard C, and CUDA C versions. Panels (a) and (g) are from Fortran versions. Panels (b) and (h) are from standard C versions. Panels (c) and (i) are from CUDA C versions. Panels (d) and (j) are the output concentration differences of Fortran and standard C versions. Panels (e) and (k) are the output concentration differences of standard C and CUDA C versions. Panels (f) and (l) are the output concentration differences of Fortran and CUDA C versions.

Figure 8. shows the boxplot of AEs and relative error (REs) in all grid boxes for the six species during the porting process. As described above, the AEs and REs introduced by the Fortran to standard C code refactoring process are significantly small, and the primary error comes from converting standard C to CUDA C. Statistically, the average of AEs (REs) of $SO_2$, $O_3$, $NO_2$, CO, $H_2O_2$ and $PSO_4$ were -0.0009 ppbV (-0.01%), 0.0004 ppbV (-0.004%), 0.0005 ppbV (0.008%), 0.03 ppbV (0.01%), $2.1 \times 10^{-5}$ ppbV (-0.01%) and 0.0002 $\mu g \cdot m^{-3}$ (0.0023%), respectively between the Fortran and CUDA C versions.

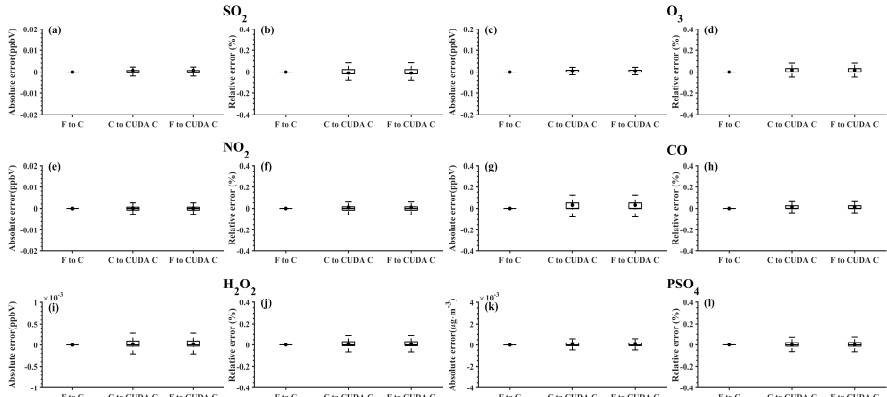

**Figure 8.** The distributions of absolute errors and relative errors for $SO_2$, $O_3$, $NO_2$, CO, $H_2O_2$ and $PSO_4$ in all of the grid boxes after 48 hours of integration.

Figure 9. presents the regionally averaged time series and AEs of $SO_2$, $O_3$, $NO_2$, CO, $H_2O_2$ and $PSO_4$. The time series between different versions is almost consistent, and the maximum AEs for above six species are 0.001ppbV, 0.005 ppbV, 0.002 ppbV, 0.03ppbV, 0.0001 ppbV and 0.0002 $\mu g \cdot m^{-3}$, respectively between the Fortran and CUDA C versions.



It is difficult to verify the scientific applicability of the results from CUDA C
version because the programming language and hardware are different between the
Fortran and CUDA C version. Here, we used the evaluation method of Wang et al.
(2021a) to compute the root mean square errors (RMSEs) of SO2, O3, NO2, CO, H2O2
and PSO4 between the Fortran and CUDA C versions, which are 0.0007 ppbV, 0.001
ppbV, 0.0002 ppbV, 0.0005 ppbV, 0.00003 ppbV, and 0.0004 $\mu g \cdot m^{-3}$ respectively,
much smaller than the spatial variation of the whole region, which is 7.0 ppbV
(approximately 0.004%), 9.7 ppbV (approximately 0.003%), 7.4 ppbV (approximately
0.003%), 142.2 ppbV (approximately 0.006%), 0.2ppbV (approximately 0.015%) and
1.7 $\mu g \cdot m^{-3}$ (approximately 0.004%). It is indicated that the bias between CUDA C
and Fortran version of the above six species is negligible compared with their own
spatial changes, and the results of the CUDA C version are generally acceptable for
research.

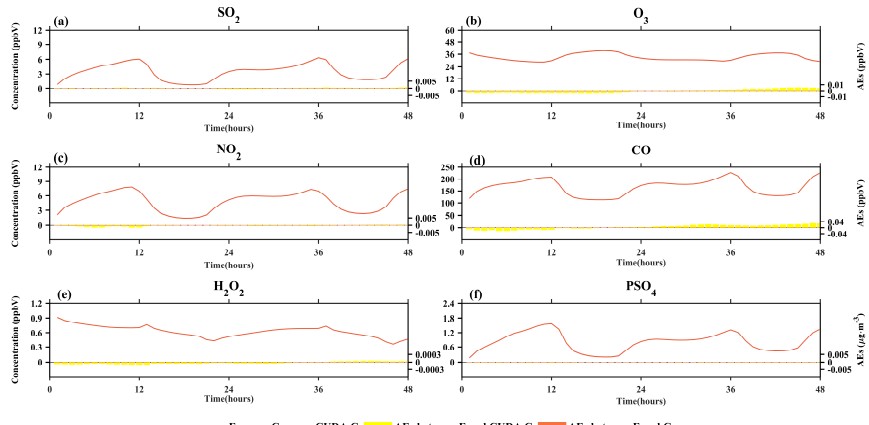


**Figure 9.** Time series and AEs of SO$_2$, O$_3$, NO$_2$, CO, H$_2$O$_2$ and PSO$_4$ outputted by CAMx model
for Fortran, standard C, and CUDA C versions.
**4.3.   Offline performance comparison of GPU-HADVPPM**
As described in the Sect. 4.2, we validate that the CAMx model result of the
CUDA C version can be generally acceptable for scientific research. We tested the
offline performance of the HADVPPM and GPU-HADVPPM scheme on 1 CPU core



and 1 GPU card, respectively. There are 7 variables input into the HADVPPM program, which are nn, dt, dx, con, vel, area, and areav, and their specific meanings are shown in Table S1.

Firstly, we use random_number function in Fortran to create random single-precision floating-point numbers of different sizes for the above 7 variables, and then transmit these random numbers to the hadvppm Fortran program and hadvppm_kernel CUDA C program for computation, respectively. Finally, test the offline performance of the HADVPPM and GPU-HADVPPM on the CPU and GPU platforms. During the offline performance experiments, we used two different CPUs and GPUs described in the Sect. 2.4., and the experimental results are shown in Figure 10.

On the CPU platform, the wall time of hadvppm Fortran program does not change significantly when the data size is less than 1000. With the increase in the data size, its wall time increases linearly. When the data size reaches $10^7$, the wall time of the hadvppm Fortran program on Intel Xeon E5-2682v4 and Intel Platinum 8168 CPU platforms is 1737.3ms and 1319.0ms, respectively. On the GPU platform, the reconstructed and extended CUDA C program implements parallel computation of multiple grid points by executing a large number of kernel function copies, so the computational efficiency of hadvppm_kernel CUDA C code on it is significantly improved. In the size of $10^7$ random numbers, the hadvppm_kernel CUDA C program takes only 12.1ms and 1.6ms to complete the computation on the NVIDIA Tesla K40m and NVIDIA Tesla V100 GPU.

Figure 10. (b) shows the speedup of HADVPPM and GPU-HADVPPM on CPU platform and GPU platform under different data sizes. When mapping the HADVPPM scheme to GPU, the computational efficiency under different data size is not only significantly improved, but also the larger the data size, the more obvious the acceleration effect of the GPU-HADVPPM. For example, in the size of $10^7$ random numbers, the GPU-HADVPPM achieved 1113.6x and 845.4x acceleration on the NVIDIA Tesla V100 GPU, respectively, compared to the two CPU platforms. Although the K40m GPU's single-card computing performance is slightly lower than that of the



V100 GPU, GPU-HADVPPM can also achieve up to 143.3x and 108.8x acceleration.

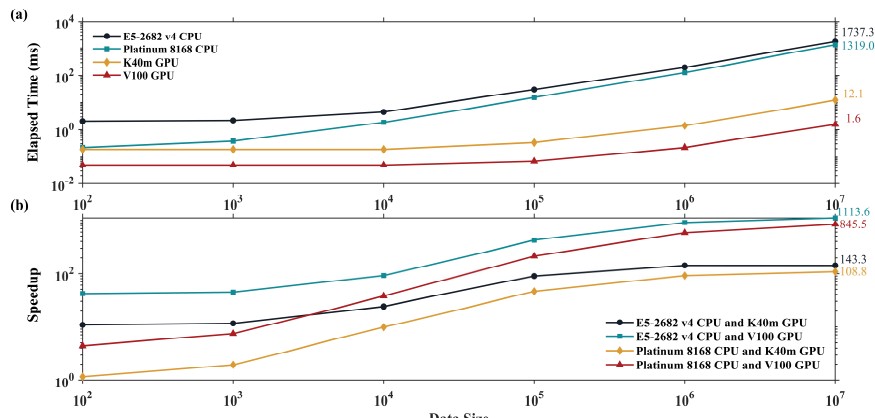

**Figure 10.** The offline performance of the HADVPPM and GPU-HADVPPM scheme on CPU and

GPU. The unit of the wall times for the offline performance experiments is millisecond(ms).

## 4.4.    Coupling performance comparison of GPU-HADVPPM with different GPU configurations

### 4.4.1.    CAMx-CUDA on a single GPU

Offline performance results show that the larger the data size, the more obvious

the acceleration effect of GPU-HADVPPM scheme. After coupling the GPU-
HADVPPM to CAMx without changing the advection module algorithm, the overall
computational efficiency of CAMx-CUDA model is extremely low, and it takes about
621 minutes to complete one-hour integration on the V100 cluster. Therefore, according
to the optimization scheme in Sect. 3.2, by optimizing the algorithm of xyadvec Fortran
program, we gradually increase the size of data transmitted and reduce the frequency
of data transmission between CPU and GPU. When the data transmission frequency
between CPU and GPU is reduced to 1 within one time-step, we further optimize the
GPU memory access order on GPU card, eliminate unnecessary assignment loops
before kernel functions launched and use thread and block indices.

Table 5. lists the total elapsed time for different versions of CAMx-CUDA model

during the optimization, as described in Section 3.2. Since the xyadvec program in the





CAMx-CUDA V1.0 is not optimized, it is extremely computationally inefficient when
starting two CPU processes and configuring a GPU card for P1. On the K40m and V100
cluster, it takes 10829 seconds and 37237 seconds respectively to complete 1-hour
simulation.
By optimizing the algorithm of xyadvec Fortran program and hadvppm_kernel
CUDA C program, the frequency of data transmission between CPU and GPU was
decreased, and the overall computing efficiency was improved after GPU-HADVPPM
coupling to CAMx-CUDA model. In CAMx-CUDA V1.2, the frequency of data
transmission between CPU-GPU within one time step is reduced to 1, and the elapsed
time on the two heterogeneous clusters is 1207 seconds and 548 seconds, respectively,
and the speedup is 9.0x and 68.0x compared to the CAMx-V1.0.
GPU memory access order can directly affect the overall computational
efficiency of GPU-HAVPPM on the GPU. In CAMx-CUDA V1.3, we have optimized
the memory access order of hadvppm_kernel CUDA C program on the GPU and
eliminated unnecessary assignment loops before kernel functions launched, which
further improved the CAMx-CUDA model computational efficiency, resulting in 12.7x
and 94.8x speedups.
Using thread and block indices to compute horizontal grid points simultaneous can
greatly improve the computational efficiency of GPU-HADVPPM and thus reduce the
overall elapsed time of CAMx-CUDA model. CAMx-CUDA V1.4 further reduces the
elapsed time by 378 seconds and 103 seconds respectively on K40m cluster and V100
cluster compared with CAMx-CUDA V1.3, and achieving up to 29.0x and 128.4x
speedup compared with CAMx-CUDA V1.0.
**Table 5.** Total elapsed time for different versions of CAMx-CUDA during the optimization. The
unit of elapsed time for experiments is seconds (s).

| Versions | K40m cluster | | V100 cluster | |
|---|---|---|---|---|
| | Elapsed Time | Speedup | Elapsed Time | Speedup |
| CAMx-CUDA V1.0 | 10829 | 1.0 | 37237 | 1.0 |



| | | | | |
|---|---|---|---|---|
| CAMx-CUDA V1.1 | 1403 | 7.7 | 1082 | 34.4 |
| CAMx-CUDA V1.2 | 1207 | 9.0 | 548 | 68.0 |
| CAMx-CUDA V1.3 | 751 | 12.7 | 393 | 94.8 |
| CAMx-CUDA V1.4 | 373 | 29.0 | 290 | 128.4 |

In terms of the single-module computational efficiency of HADVPPM and GPU-
HADVPPM, we further test their performance in CPU and GPU using system_clock
functions in the Fortran language and cudaEvent_t in CUDA programming. Figure 11.
show the elapsed time of HADVPPM and GPU-HADVPPM in CAMX-CUDA V1.4
(GPU-HADVPPM V1.4) on K40m cluster and V100 cluster. On K40m cluster, it takes
37.7 seconds and 29.6 seconds to launch the Intel Xeon E5-2682 v4 CPU and a NVIDIA
Tesla K40m GPU to run HADVPPM and GPU-HADVPPM, respectively, with 1.3x
acceleration. On the V100 cluster, it takes 30.6 seconds to launch the Intel Xeon
Platinum 8168 CPU to complete the HADVPPM operation, and only 1.6 seconds to run
the GPU-HADVPPM using a NVIDIA V100 GPU after porting, with a speedup of
19.4x.

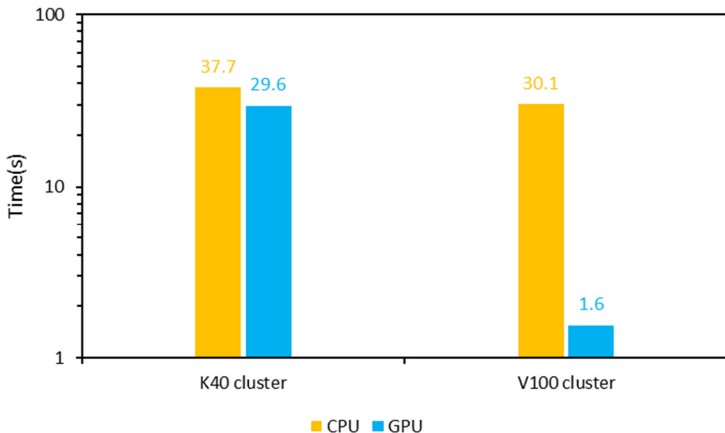


**Figure 11.** The elapsed time of HADVPPM and GPU-HADVPPM V1.4 on CPU and GPU. The
unit is seconds (s).



### 4.4.2. CAMx-CUDA on multiple GPUs

To make full use of multi-core and multi-GPU in the heterogeneous cluster, MPI+CUDA acceleration algorithm was implemented to improve the total computational performance of the CAMx-CUDA model. Two different compile flags were implemented in this study before comparing the computational efficiency of CAMx-CUDA V1.3 and V1.4 on multiple GPUs, namely *-mieee-fp* and *-fp-model precise*. The *-mieee-fp* compile flag comes from the *Makefile* of the official CAMx version, which uses the IEEE standard to compare floating-point numbers. Its computation accuracy is higher, but the efficiency is slower. The *-fp-model precise* compile flag control the balance between precision and efficiency of floating-point calculations, and it can force the compiler to use the vectorization of some calculations under the value-safe. The experiment results show that *-fp model precise* compile flag is 41.4% faster than *-mieee-fp*, and the AEs of the simulation results are less than $\pm 0.05$ppbV (Figure S2). Therefore, the *-fp model precise* compile flag is implemented when comparing the computational efficiency of CAMx-CUDA V1.3 and V1.4 on multiple GPU cards. Figure 12. shows the total elapsed time and speedup of CAMx-CUDA V1.3 and V1.4 on the V100 cluster. The total elapsed time decreases as the number of CPU cores and GPU cards increases. When starting 8 CPU cores and 8 GPU cards, the speedup of CAMx-CUDA V1.4 is increased from 3.9x to 4.5x compared with V1.3, and the computational efficiency is increased by 35.0%.



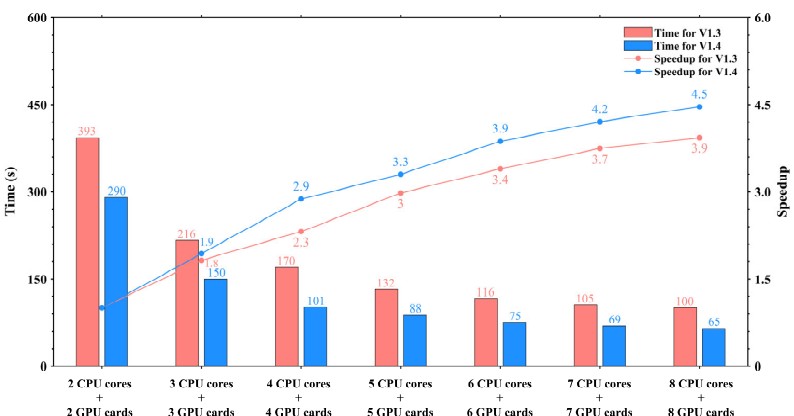

**Figure 12.** The total elapsed time and speedup of CAMx-CUDA V1.3 and V1.4 on multiple

GPUs. The unit of elapsed time for experiments is seconds (s).

## 5.  Conclusions and discussion

  GPU accelerators are playing an increasingly important role in high-performance

computing. In this study, a GPU acceleration version of the PPM solver (GPU-

HADVPPM) of horizontal advection for air quality model is developed, that can be run

on GPU accelerators using the standard C programming language and CUDA

technology. Offline performance experiments results show that K40m and V100 GPU

can achieve up to 845.4x and 1113.6x speedup, respectively, and the larger the data

input to the GPU, the more obvious the acceleration effect. After coupling GPU-

HADVPPM to CAMx model, a series of optimization measures are taken, including

reducing the CPU-GPU communication frequency, increasing the size of data

computation on GPU, optimizing the GPU memory access order, and using thread and

block indices to improve the overall computing performance of CAMx-CUDA model.

Using a single GPU card, the optimized CAMx-CUDA V1.4 model improves the

computing efficiency by 29.0x and 128.4x on the K40m cluster and the V100 cluster,

respectively. In terms of the single-module computational efficiency of GPU-



HADVPPM, it can achieve 1.3x and 19.4x speedup on NVIDIA Tesla K40m GPU and
NVIDA Tesla V100 GPU respectively. To make full use of multi-core and multi-GPU
supercomputers and further improve the total computational performance of CAMx-
CUDA model, a parallel architecture with an MPI+CUDA hybrid paradigm is presented.
After implementing the acceleration algorithm, the total elapsed time decreases as the
number of CPU cores and GPU cards increases, and it can achieve up to 4.5x speedup
when launch 8 CPU cores and 8 GPU cards compared with 2 CPU cores and 2 GPU
cards.
The communication bandwidth of data transfer is one of the main issues for
restricting the computing performance of CUDA C codes on GPUs. This restriction not
only holds true for GPU-HADVPPM, but also WRF module as well (Mielikainen et al.,
2012b; Mielikainen et al., 2013b; Huang et al., 2013). Data transfer efficiency between
CPU and GPU can be optimized.
The results of this offline performance experiment shows that the larger the
amount of data transferred to the GPU, the more obvious the acceleration effect.
However, the number of 3D grids points in the coupling test case in this paper is only
145×157×14, a larger simulation case can be used.
The computation of HADVPPM is just a small part of the whole CAMx model.
When CAMx model will be completely implemented on GPU, the inputs for GPU-
HADVPPM do not have to be transferred from CPU. Similarly, outputs of GPU-
HADVPPM will be directly inputs to another CAMx module on GPU. Therefore, the
role of I/O is greatly diminished once all of CAMx model have been converted to run
on GPUs. In the future, other CAMx modules can be considered to adopt the scheme
given in this paper to carry out heterogeneous porting.

*Code and data availability.* The source codes of the CAMx version 6.10 are available
at https://camx-wp.azurewebsites.net/download/source/ (last access: 24 March 2023,
ENVIRON,2022). The dataset related to this paper and CAMx-CUDA codes are
available online via ZENODO (http://doi.org/10.5281/zenodo.7765218; Cao et



al.,2023).

**Acknowledgements.**    The National Key R&D Program of China (2020YFA0607804
& 2017YFC0209805), and National Supercomputing Center in Zhengzhou Innovation
Ecosystem Construction Technology Special Program (Grant No.201400210700), and
Beijing Advanced Innovation Program for Land Surface funded this work. The research
is support by the High Performance Scientific Computing Center (HSCC) of Beijing
Normal University and the National Supercomputing Center in Zhengzhou.

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
