# Peer review of "GPU-HADVPPM V1.0: A high-efficiency parallel GPU design of"

_EGUsphere, 2023_

## Community Comment (CC1)

This paper presents implementation and optimization of air quality model CAMx using Cuda C targeting GPU clusters. Experiment results show that GPU-HADVPPM can achieve about 1000x acceleration, the series of optimization can acheive about dozens of times acceleration and the final version of CAMx-GPU can achieve 4.5x speedup with 8 CPU cores and 8 GPU accelerators on V100 cluster. Here are some specific comments.

Response: We appreciate the editor for reviewing our manuscript and for the valuable suggestions, which we will address point by point in the following.

1. This paper need to be more convincing about how some experiment results will be explained. Such as: the offline experiment achieves about 1000x acceleration which is far exceed the ratio of theoretical peak performance of GPU and CPU.

Response: Thanks for the constructive comment. In the thread hierarchy, the thread is the most basic unit of GPU for parallel computing. Threads can be organized into one-dimensional, two-dimensional, or three-dimensional blocks. In a three-dimensional block, each dimension can contain a maximum number of threads of 1024, 1024, and 64, respectively. Similarly, blocks can be organized into one-dimensional, two-dimensional, or three-dimensional grids. In a three-dimensional grid, each dimension can contain a maximum number of blocks of $2^{31} - 1$, 65535, and 65535, respectively. It is theoretically possible to distribute a large number of copies of kernel functions into tens of billions of threads for parallel computing without exceeding the GPU memory. In the offline performance test, we adopted a one-dimensional thread organization, with 1024 threads per block and several blocks based on the array size. For example, in the data size of $10^7$, the GPU can implement the parallel computation of 10 million threads, while the CPU can only use serial cyclic computation. Therefore, the offline experiments achieve up to 1100x acceleration, and the larger the data size, the more pronounced the advantage of GPU multithreading architecture and the higher the computing efficiency. In addition, the data transfer time between CPU and GPU is not considered in the offline performance experiments. Also, without I/O, the GPU-based SBU-YLIN scheme in the WRF model can achieve **896x speedup** compared to a Fortran implementation running on a CPU (Mielikainen et al.,2012). We have revised this part in lines 463-474, which are as follows:

As described in Sect. 3.2, the thread is the most basic unit of GPU for parallel computing. Each dimension of the three-dimensional block can contain a maximum number of threads of 1024,1024, and 64, respectively. Each dimension of the three-dimensional grid can contain a maximum number of blocks of $2^{31} - 1$, 65535, and 65535. It is theoretically possible to distribute a large number of copies of kernel functions into tens of billions of threads for parallel computing without exceeding the GPU memory. In the offline performance experiments, the GPU achieved up to 10 million threads of parallel computing, while the CPU can only use serial cyclic computation. Therefore, GPU-HADVPPM achieves a maximum acceleration of about 1100x without I/O. In addition to this study, the GPU-based SBU-YLIN scheme in the WRF model can achieve 896x acceleration compared to the Fortran implementation running on the CPU (Mielikainen et al., 2012b).

2. It is suggested to supplement performance data of CAMx implemented by Fortran and C in section4.4.1.

Response: Thanks for the constructive comment. We supplement the coupling performance experiment results for the C version of HADVPPM. On the K40m and V100 cluster, the elapsed time of the C version HADVPPM is 51.4 seconds and 45.2 seconds, respectively, 26.7% and 33.4% lower than that of the Fortran version HADVPPM. We have revised this part in lines 519-534, which are as follows:

In terms of the single module computational efficiency of HADVPPM and GPU-HADVPPM, we further coupling test the computational performance of the Fortran version HADVPPM on the CPU, C version HADVPPM on the CPU, and CUDA C version GPU-HADVPPM in CAMx-CUDA V1.4 (GPU-HADVPPM V1.4) on the GPU, using system_clock functions in the Fortran language and cudaEvent_t in CUDA programming. The specific results are shown in Figure 11. On the K40m cluster, it takes 37.7 seconds and 51.4 seconds to launch the Intel Xeon E5-2682 v4 CPU to run Fortran and C version HADVPPM, the C version is 26.7% slower than the Fortran version. After the CUDA technology was used to convert the C code into CUDA C, the CUDA C version took 29.6 seconds to launch an NVIDIA Telsa K40m GPU to run GPU-HADVPPM V1.4, with 1.3x and 1.7x acceleration. On the V100 cluster, the Fortran, the C, and the CUDA C version are computationally more efficient than those on the K40m cluster. It takes 30.1 seconds and 45.2 seconds to launch Intel Xeon Platinum 8168 CPU to run Fortran and C version HADVPPM and 1.6

seconds to run the GPU-HADVPPM V1.4 using an NVIDIA V100 GPU. The computational efficiency of the CUDA C version is 18.8x and 28.3x higher than Fortran and C versions.

[Figure]

Figure 11. The elapsed time of the Fortran version HADVPPM on the CPU, the C version HADVPPM on the CPU, and CUDA C version GPU-HADVPPM V1.4 on the GPU. The unit is seconds (s).

3. There are some typing errors, such as CAS-EMS should be CAS-ESM, CAMx-V1.0 in line 492 should be CAMx-CUDA V1.0.

Response: Sorry for this mistake. We have corrected typing errors in line 64, 84, and 504, which are as follows:

Line 64: The GPU has proven successful in weather models such as Non-Hydrostatic Icosahedral Model (NIM; Govett et al.,2017), Global/Regional Assimilation and Prediction System (GRAPES; Xiao et al., 2022), and Weather Research and Forecasting model (WRF; Huang et al., 2011; Huang et al., 2012; Mielikainen et al., 2012a; Mielikainen et al., 2012b; Mielikainen et al., 2013a ; Mielikainen et al., 2013b; Price et al., 2014; Huang et al., 2015), ocean models such as LASG/IAP Climate System Ocean Model (LICOM; Jiang et al., 2019; Wang et al., 2021a) and Princeton Ocean Model (POM; Xu et al., 2015), and the Earth System Model of Chinese Academy of Sciences (**CAS-ESM**; Wang et al., 2021b ; Wang et al., 2021c).

Line 84: In terms of climate system model, Wang et al., (2021c) and Wang et al., (2021b) used CUDA Fortran and CUDA C to carry out heterogeneous porting of the RRTMG_SW and RRTMG_LW scheme of the atmospheric component model of the **CAS-ESM** earth system model, and achieved a 38.88x and 77.78x acceleration respectively.

Line 504: In CAMx-CUDA V1.2, the frequency of data transmission between CPU-GPU within one time step is reduced to 1, and the elapsed time on the two heterogeneous clusters is 1207 seconds and 548 seconds, respectively, and the speedup is 9.0x and 68.0x compared to the **CAMx-CUDA V1.0**.

Reference

Mielikainen, J., Huang, B., Huang, H.-L. A., and Goldberg, M. D.: GPU Implementation of Stony Brook University 5-Class Cloud Microphysics Scheme in the WRF, IEEE Journal of Selected Topics in Applied Earth Observations and Remote Sensing, 5, 625-633, 10.1109/jstars.2011.2175707, 2012.

---

## Community Comment (CC2)

GPU accelerators have great potential to improve the efficiency of air quality modeling. The authors have explained the steps taken to convert a PPM solver of horizontal advection for air quality model CAMx into a new Compute Unified Device Architecture C (CUDA C) code and how this optimization approach has resulted in significant improvement in computational efficiency. This paper presents a promising approach for using GPU accelerators in air quality modeling and provides valuable insights into the optimization measures that can be taken to improve the overall computing performance of model. The authors' findings can potentially benefit researchers in air quality modeling and related fields. My comments listed below.

Response: We appreciate the editor for reviewing our manuscript and for the valuable suggestions, which we will address point by point in the following.

1. The manuscript should be written more concisely, removing unnecessary material. For example, the information presented in Table 1 partly duplicates that presented in Figure 1b. Figure 9 doesn't provide too much important information and can be moved to Supplementary.

   Response: Thanks for the constructive comment. We have removed Table 1, and placed Figure 9 in the supplementary. The relevant expressions and contents in the paper have also been modified in lines 156-162 and 417-421, which are as follows:

   Lines 156-162: The VTune tool detects each module's runtime and the most time-consuming functions on P1. As shown in Figure 1b, the top four time-consuming modules are chemistry, diffusion, horizontal advection, and vertical advection in the CAMx model. In the above four modules, the top five most time-consuming programs are ebirate, hadvppm, tridiag, diffus, and ebisolv programs, and the total runtime of P1 is 325.1 seconds. Top1 and Top2's most time-consuming programs take 49.4 and 35.6 seconds, respectively.

   Lines 417-421: In terms of time series, the regionally averaged time series of the three versions are almost consistent (as is shown in Figure S2), and the maximum AEs for the above six species are 0.001ppbV, 0.005 ppbV, 0.002 ppbV, 0.03ppbV, 0.0001 ppbV and 0.0002 $\mu g \cdot m^{-3}$, respectively, between the Fortran and CUDA C versions.

2. The authors select the hadvppm program to implement the heterogeneous porting. It could benefit from providing some additional context about the motivation for selecting hadvppm

program.

Response: Thanks for the constructive comment. There are some reasons for selecting hadvppm for heterogeneous porting. Firstly, the advection module is one of the compulsory modules of the air quality model, which is mainly used to simulate the transport process of air pollutants, and it is also a hotspot module detected by the Intel VTune tool. Then, typical air quality models CAMx, CMAQ, and NAQPMS include advection modules and use the exact PPM advection solver. The heterogeneous version developed in this study can be directly applied to the above models. Furthermore, the weather model (e.g., WRF) also contains an advection module, so this study's heterogeneous porting method and experience can be used for reference. We have revised this part in lines 163-173, which are as follows:

By consideration, the hadvppm program was selected to carry out heterogeneous porting for some reasons. Firstly, the advection module is one of the compulsory modules of the air quality model, which is mainly used to simulate the transport process of air pollutants, and it is also a hotspot module detected by the Intel VTune tool. Then, typical air quality models CAMx, CMAQ, and NAQPMS include advection modules and use the exact PPM advection solver. The heterogeneous version developed in this study can be directly applied to the above models. Furthermore, the weather model (e.g., WRF) also contains an advection module, so this study's heterogeneous porting method and experience can be used for reference. Therefore, a GPU acceleration version of the HADVPPM scheme, namely GPU-HADVPPM, is built to improve CAMx performance.

3.  In Figure 5-7, large absolute errors mainly occur over Zhangjiakou City for $SO_2$, $NO_2$ CO and $PSO_4$ while simulated $O_3$ and $H_2O_2$ show large errors over Baoding city. Please explain how do porting process cause such patterns.

Response: Thanks for the constructive comment. During the process of heterogeneous porting, the CUDA technology was used to convert the standard C code into CUDA C to make the hadvppm program computable on the GPU. However, due to the slight difference in data operation and accuracy between CPU and GPU(NVIDIA,2023), the concentration variable of hadvppm program appears to have minimal negative values (about $-10^{-4} \sim -10^{-9}$) when integrating on GPU. In order to allow the program to continue running, we forcibly replace these negative values with $10^{-9}$. It is because these negative values are replaced by positive

values that the simulation results are biased. Furthermore, we adopted the evaluation method of Wang et al. (2021) to compute the ratio between the root-mean-square-error and standard deviation of six species, such as $SO_2$, $O_3$, $NO_2$, $CO$, $H_2O_2$, and $PSO_4$. The results were 0.004%, 0.003%, 0.003%, 0.006%, 0.015%, and 0.004%, respectively, which were far less than the 0.2% mentioned by Wang et al. (2021), so the absolute errors of simulation results in this study are within the acceptable range. We have revised this part in lines 375-383, which are as follows:

During the porting process, the primary error came from converting standard C to CUDA C, and the main reason was the hardware difference between the CPU and GPU. Due to the slight difference in data operation and accuracy between CPU and GPU(NVIDIA,2023), the concentration variable of hadvppm program appears to have minimal negative values (about -$10^{-4}$~-$10^{-9}$) when integrating on GPU. In order to allow the program to continue running, we forcibly replace these negative values with $10^{-9}$. It is because these negative values are replaced by positive values that the simulation results are biased.

4. I'd suggest the authors discussing in the Conclusions and Discussion section any limitations of the current approach, which may warrant future refinement.

Response: Thanks for the constructive comment. There are some limitations of the current approach.

1) We currently implemented thread and block co-indexing to compute horizontal grid points in parallel. Given the CAMx model 3-dimensional grid computing characteristics, 3-dimensional thread and block co-indexing will be considered to compute 3-dimensional grid points in parallel.

2) The communication bandwidth of data transfer is one of the main issues for restricting the computing performance of CUDA C codes on GPUs. In this study, data transmission efficiency between CPU and GPU is improved only by reducing communication frequency. In the future, more technologies, such as pinned memory (Wang et al.,2016), will be considered to solve the communication bottleneck between CPU and GPU.

3) In order to further improve the overall computational efficiency of the CAMx model, the heterogeneous porting scheme proposed in this study will be considered to carry out the heterogeneous porting of other CAMx modules in the future.

We have added this part in lines 600-614, which are as follows:

However, there are some limitations of the current approach which are as follows:

1) We currently implemented thread and block co-indexing to compute horizontal grid points in parallel. Given the CAMx model 3-dimensional grid computing characteristics, 3-dimensional thread and block co-indexing will be considered to compute 3-dimensional grid points in parallel.

2) The communication bandwidth of data transfer is one of the main issues for restricting the computing performance of CUDA C codes on GPUs. This restriction not only holds true for GPU-HADVPPM, but also WRF module as well (Mielikainen et al., 2012b; Mielikainen et al., 2013b; Huang et al., 2013). In this study, data transmission efficiency between CPU and GPU is improved only by reducing communication frequency. In the future, more technologies, such as pinned memory (Wang et al.,2016), will be considered to solve the communication bottleneck between CPU and GPU.

3) In order to further improve the overall computational efficiency of the CAMx model, the heterogeneous porting scheme proposed in this study will be considered to carry out the heterogeneous porting of other CAMx modules in the future.

5. Line 173-177, 337-342: Please unify the tense of the sentence.

Response: Sorry for these mistakes. We have revised this part in lines 180-187 and 342-349, which are as follows:

Lines 180-187: The heterogeneous scheme of CAMx-CUDA is shown in Figure 2. The second time-consuming hadvppm program in the CAMx model was selected to implement the heterogeneous porting. In order to map the hadvppm program to the GPU, the Fortran code was converted to standard C code. Then, CUDA programing language, which was tailor-made for NVIDIA, was added to convert the standard C code into CUDA C for data-parallel execution on GPU, as GPU-HADVPPM. It prepared the input data for GPU-HADVPPM by constructing random numbers and tested its offline performance on the GPU platform.

Lines 342-349: The validation and evaluation of porting the HADVPPM scheme from the CPU to the GPU platform were conducted using offline and coupling performance experiments. First, we validated the result between different CAMx versions, and then the offline performance of the GPU-HADVPPM on a single GPU was tested by offline experiment. Finally, the coupling performance experiments illustrate its potential in three dimensions with varying chemical

regimes. Sect.4.2 and Sect.4.4, the CAMx version of the HADVPPM scheme written by Fortran language, standard C, and CUDA C, is named F, C, and CUDA C, respectively.

6.    Line 354: Please clarify "30-min spatial resolution".

Response: Sorry for the confusion. Minute is the angular resolution. According to the conversion formula, 30-min is 0.5 degrees, and the corresponding range resolution is about 55.6 kilometers at mid-latitude. We have revised this part in line 361, which are as follows:

>    30-min (about 55.6km at mid-latitude) spatial resolution

Reference

NVIDIA: Floating Point and IEEE 754 Compliance for NVIDIA GPUs. Release 12.1, available at: https://docs.nvidia.com/cuda/floating-point/#differences-from-x86 (last access: 18 May 2023), 2023.

Wang, P., Jiang, J., Lin, P., Ding, M., Wei, J., Zhang, F., Zhao, L., Li, Y., Yu, Z., Zheng, W., Yu, Y., Chi, X., and Liu, H.: The GPU version of LASG/IAP Climate System Ocean Model version 3 (LICOM3) under the heterogeneous-compute interface for portability (HIP) framework and its large-scale application, Geosci. Model Dev., 14, 2781-2799, 10.5194/gmd-14-2781-2021, 2021.

Wang, Z., Wang, Y., Wang, X., Li, F., Zhou, C., Hu, H., and Jiang, J.: GPU-RRTMG_SW: Accelerating a Shortwave Radiative Transfer Scheme on GPU, IEEE Access, 9, 84231-84240, 10.1109/access.2021.3087507, 2016.

---

## Author Comment (AC1)

**Response to Reviewers' comments**

We are thankful to the two reviewers for their thoughtful and constructive comments that help us improve the manuscript substantially. We have revised the manuscript accordingly. Listed below is our point-to-point response in blue to each comment that was offered by the reviewers.

**Response to Reviewer #1**

General comments:

This paper presents implementation and optimization of air quality model CAMx using Cuda C targeting GPU clusters. Experiment results show that GPU-HADVPPM can achieve about 1000x acceleration, the series of optimization can acheive about dozens of times acceleration and the final version of CAMx-GPU can achieve 4.5x speedup with 8 CPU cores and 8 GPU accelerators on V100 cluster. Here are some specific comments.

Response: We appreciate the editor for reviewing our manuscript and for the valuable suggestions, which we will address point by point in the following.

1.  This paper need to be more convincing about how some experiment results will be explained. Such as: the offline experiment achieves about 1000x acceleration which is far exceed the ratio of theoretical peak performance of GPU and CPU.

    Response: Thanks for the constructive comment. In the thread hierarchy, the thread is the most basic unit of GPU for parallel computing. Threads can be organized into one-dimensional, two-dimensional, or three-dimensional blocks. In a three-dimensional block, each dimension can contain a maximum number of threads of 1024, 1024, and 64, respectively. Similarly, blocks can be organized into one-dimensional, two-dimensional, or three-dimensional grids. In a three-dimensional grid, each dimension can contain a maximum number of blocks of $2^{31} - 1, 65535$, and 65535, respectively. It is theoretically possible to distribute a large number of copies of kernel functions into tens of billions of threads for parallel computing without exceeding the GPU memory. In the offline performance test, we adopted a one-dimensional thread organization, with 1024 threads per block and several blocks based on the array size. For example, in the data size of $10^7$, the GPU can implement the parallel computation of 10 million

threads, while the CPU can only use serial cyclic computation. Therefore, the offline experiments achieve up to 1100x acceleration, and the larger the data size, the more pronounced the advantage of GPU multithreading architecture and the higher the computing efficiency. In addition, the data transfer time between CPU and GPU is not considered in the offline performance experiments. Also, without I/O, the GPU-based SBU-YLIN scheme in the WRF model can achieve **896x speedup** compared to a Fortran implementation running on a CPU (Mielikainen et al.,2012). We have revised this part in lines 478-489, which are as follows:

*As described in Sect. 3.2, the thread is the most basic unit of GPU for parallel computing. Each dimension of the three-dimensional block can contain a maximum number of threads of 1024,1024, and 64, respectively. Each dimension of the three-dimensional grid can contain a maximum number of blocks of $2^{31} - 1$, 65535, and 65535. It is theoretically possible to distribute a large number of copies of kernel functions into tens of billions of threads for parallel computing without exceeding the GPU memory. In the offline performance experiments, the GPU achieved up to 10 million threads of parallel computing, while the CPU can only use serial cyclic computation. Therefore, GPU-HADVPPM achieves a maximum acceleration of about 1100x without I/O. In addition to this study, the GPU-based SBU-YLIN scheme in the WRF model can achieve 896x acceleration compared to the Fortran implementation running on the CPU (Mielikainen et al., 2012b).*

2. It is suggested to supplement performance data of CAMx implemented by Fortran and C in section4.4.1.

Response: Thanks for the constructive comment. We supplement the coupling performance experiment results for the C version of HADVPPM. On the K40m and V100 cluster, the elapsed time of the C version HADVPPM is 51.4 seconds and 45.2 seconds, respectively, 26.7% and 33.4% lower than that of the Fortran version HADVPPM. We have revised this part in lines 534-549, which are as follows:

*In terms of the single module computational efficiency of HADVPPM and GPU-HADVPPM, we further coupling test the computational performance of the Fortran version HADVPPM on*

*the CPU, C version HADVPPM on the CPU, and CUDA C version GPU-HADVPPM in CAMx-CUDA V1.4 (GPU-HADVPPM V1.4) on the GPU, using system_clock functions in the Fortran language and cudaEvent_t in CUDA programming. The specific results are shown in Figure 11. On the K40m cluster, it takes 37.7 seconds and 51.4 seconds to launch the Intel Xeon E5-2682 v4 CPU to run Fortran and C version HADVPPM, the C version is 26.7% slower than the Fortran version. After the CUDA technology was used to convert the C code into CUDA C, the CUDA C version took 29.6 seconds to launch an NVIDIA Telsa K40m GPU to run GPU-HADVPPM V1.4, with 1.3x and 1.7x acceleration. On the V100 cluster, the Fortran, the C, and the CUDA C version are computationally more efficient than those on the K40m cluster. It takes 30.1 seconds and 45.2 seconds to launch Intel Xeon Platinum 8168 CPU to run Fortran and C version HADVPPM and 1.6 seconds to run the GPU-HADVPPM V1.4 using an NVIDIA V100 GPU. The computational efficiency of the CUDA C version is 18.8x and 28.3x higher than Fortran and C versions.*

[Figure]

*Figure 10. The elapsed time of the Fortran version HADVPPM on the CPU, the C version HADVPPM on the CPU, and CUDA C version GPU-HADVPPM V1.4 on the GPU. The unit is seconds (s).*

3. There are some typing errors, such as CAS-EMS should be CAS-ESM, CAMx-V1.0 in line 492 should be CAMx-CUDA V1.0.

Response: Sorry for this mistake. We have corrected typing errors in line 67, 87, and 519, which are as follows:

*Line 67: The GPU has proven successful in weather models such as Non-Hydrostatic Icosahedral Model (NIM; Govett et al.,2017), Global/Regional Assimilation and Prediction System (GRAPES; Xiao et al., 2022), and Weather Research and Forecasting model (WRF; Huang et al., 2011; Huang et al., 2012; Mielikainen et al., 2012a; Mielikainen et al., 2012b; Mielikainen et al., 2013a ; Mielikainen et al., 2013b; Price et al., 2014; Huang et al., 2015), ocean models such as LASG/IAP Climate System Ocean Model (LICOM; Jiang et al., 2019; Wang et al., 2021a) and Princeton Ocean Model (POM; Xu et al., 2015), and the Earth System Model of Chinese Academy of Sciences (**CAS-ESM**; Wang et al., 2016 ; Wang et al., 2021b).*

*Line 87: In terms of climate system model, Wang et al., (2021c) and Wang et al., (2021b) used CUDA Fortran and CUDA C to carry out heterogeneous porting of the RRTMG_SW and RRTMG_LW scheme of the atmospheric component model of the **CAS-ESM** earth system model, and achieved a 38.88x and 77.78x acceleration respectively.*

*Line 519: In CAMx-CUDA V1.2, the frequency of data transmission between CPU-GPU within one time step is reduced to 1, and the elapsed time on the two heterogeneous clusters is 1207 seconds and 548 seconds, respectively, and the speedup is 9.0x and 68.0x compared to the **CAMx-CUDA V1.0**.*

**Response to Reviewer #2**

GPU accelerators have great potential to improve the efficiency of air quality modeling. The authors have explained the steps taken to convert a PPM solver of horizontal advection for air quality model CAMx into a new Compute Unified Device Architecture C (CUDA C) code and how this optimization approach has resulted in significant improvement in computational efficiency. This paper presents a promising approach for using GPU accelerators in air quality modeling and provides valuable insights into the optimization measures that can be taken to improve the overall computing performance of model. The authors' findings can potentially benefit researchers in air

quality modeling and related fields. My comments listed below.

Response: We appreciate the editor for reviewing our manuscript and for the valuable suggestions, which we will address point by point in the following.

1. The manuscript should be written more concisely, removing unnecessary material. For example, the information presented in Table 1 partly duplicates that presented in Figure 1b. Figure 9 doesn't provide too much important information and can be moved to Supplementary.

   Response: Thanks for the constructive comment. We have removed Table 1, and placed Figure 9 in the supplementary. The relevant expressions and contents in the paper have also been modified in lines 156-162 and 417-421, which are as follows:

   *Lines 156-162: The VTune tool detects each module's runtime and the most time-consuming functions on P1. As shown in Figure 1b, the top four time-consuming modules are chemistry, diffusion, horizontal advection, and vertical advection in the CAMx model. In the above four modules, the top five most time-consuming programs are ebirate, hadvppm, tridiag, diffus, and ebisolv programs, and the total runtime of P1 is 325.1 seconds. Top1 and Top2's most time-consuming programs take 49.4 and 35.6 seconds, respectively.*

   *Lines 417-421: In terms of time series, the regionally averaged time series of the three versions are almost consistent (as is shown in Figure S2), and the maximum AEs for the above six species are 0.001ppbV, 0.005 ppbV, 0.002 ppbV, 0.03ppbV, 0.0001 ppbV and 0.0002 $\mu g \cdot m^{-3}$, respectively, between the Fortran and CUDA C versions.*

2. The authors select the hadvppm program to implement the heterogeneous porting. It could benefit from providing some additional context about the motivation for selecting hadvppm program.

   Response: Thanks for the constructive comment. There are some reasons for selecting hadvppm for heterogeneous porting. Firstly, the advection module is one of the compulsory modules of the air quality model, which is mainly used to simulate the transport process of air

pollutants, and it is also a hotspot module detected by the Intel VTune tool. Then, typical air quality models CAMx, CMAQ, and NAQPMS include advection modules and use the exact PPM advection solver. The heterogeneous version developed in this study can be directly applied to the above models. Furthermore, the weather model (e.g., WRF) also contains an advection module, so this study's heterogeneous porting method and experience can be used for reference. We have revised this part in lines 163-173, which are as follows:

*By consideration, the hadvppm program was selected to carry out heterogeneous porting for some reasons. Firstly, the advection module is one of the compulsory modules of the air quality model, which is mainly used to simulate the transport process of air pollutants, and it is also a hotspot module detected by the Intel VTune tool. Then, typical air quality models CAMx, CMAQ, and NAQPMS include advection modules and use the exact PPM advection solver. The heterogeneous version developed in this study can be directly applied to the above models. Furthermore, the weather model (e.g., WRF) also contains an advection module, so this study's heterogeneous porting method and experience can be used for reference. Therefore, a GPU acceleration version of the HADVPPM scheme, namely GPU-HADVPPM, is built to improve CAMx performance.*

3.  In Figure 5-7, large absolute errors mainly occur over Zhangjiakou City for $SO_2$, $NO_2$ CO and $PSO_4$ while simulated $O_3$ and $H_2O_2$ show large errors over Baoding city. Please explain how do porting process cause such patterns.

    Response: Thanks for the constructive comment. During the process of heterogeneous porting, the CUDA technology was used to convert the standard C code into CUDA C to make the hadvppm program computable on the GPU. However, due to the slight difference in data operation and accuracy between CPU and GPU(NVIDIA,2023), the concentration variable of hadvppm program appears to have minimal negative values (about $-10^{-4}$~$-10^{-9}$) when integrating on GPU. In order to allow the program to continue running, we forcibly replace these negative values with $10^{-9}$. It is because these negative values are replaced by positive values that the simulation results are biased. Furthermore, we adopted the evaluation method of Wang et al. (2021) to compute the ratio between the root-mean-square-error and standard

deviation of six species, such as $SO_2$, $O_3$, $NO_2$, CO, $H_2O_2$, and $PSO_4$. The results were 0.004%, 0.003%, 0.003%, 0.006%, 0.015%, and 0.004%, respectively, which were far less than the 0.2% mentioned by Wang et al. (2021), so the absolute errors of simulation results in this study are within the acceptable range. We have revised this part in lines 375-383, which are as follows:

*During the porting process, the primary error came from converting standard C to CUDA C, and the main reason was the hardware difference between the CPU and GPU. Due to the slight difference in data operation and accuracy between CPU and GPU(NVIDIA,2023), the concentration variable of hadvppm program appears to have minimal negative values (about $-10^{-4} \sim -10^{-9}$) when integrating on GPU. In order to allow the program to continue running, we forcibly replace these negative values with $10^{-9}$. It is because these negative values are replaced by positive values that the simulation results are biased.*

4.  I'd suggest the authors discussing in the Conclusions and Discussion section any limitations of the current approach, which may warrant future refinement.

Response: Thanks for the constructive comment. There are some limitations of the current approach.

1) We currently implemented thread and block co-indexing to compute horizontal grid points in parallel. Given the CAMx model 3-dimensional grid computing characteristics, 3-dimensional thread and block co-indexing will be considered to compute 3-dimensional grid points in parallel.

2) The communication bandwidth of data transfer is one of the main issues for restricting the computing performance of CUDA C codes on GPUs. In this study, data transmission efficiency between CPU and GPU is improved only by reducing communication frequency. In the future, more technologies, such as pinned memory (Wang et al.,2016), will be considered to solve the communication bottleneck between CPU and GPU.

3) In order to further improve the overall computational efficiency of the CAMx model, the heterogeneous porting scheme proposed in this study will be considered to carry out the heterogeneous porting of other CAMx modules in the future.

We have added this part in lines 600-614, which are as follows:

*However, there are some limitations of the current approach which are as follows:*

1) *We currently implemented thread and block co-indexing to compute horizontal grid points in parallel. Given the CAMx model 3-dimensional grid computing characteristics, 3-dimensional thread and block co-indexing will be considered to compute 3-dimensional grid points in parallel.*

2) *The communication bandwidth of data transfer is one of the main issues for restricting the computing performance of CUDA C codes on GPUs. This restriction not only holds true for GPU-HADVPPM, but also WRF module as well (Mielikainen et al., 2012b; Mielikainen et al., 2013b; Huang et al., 2013). In this study, data transmission efficiency between CPU and GPU is improved only by reducing communication frequency. In the future, more technologies, such as pinned memory (Wang et al.,2016), will be considered to solve the communication bottleneck between CPU and GPU.*

3) *In order to further improve the overall computational efficiency of the CAMx model, the heterogeneous porting scheme proposed in this study will be considered to carry out the heterogeneous porting of other CAMx modules in the future.*

5. Line 173-177, 337-342: Please unify the tense of the sentence.

Response: Sorry for these mistakes. We have revised this part in lines 180-187 and 342-349, which are as follows:

*Lines 180-187: The heterogeneous scheme of CAMx-CUDA is shown in Figure 2. The second time-consuming hadvppm program in the CAMx model was selected to implement the heterogeneous porting. In order to map the hadvppm program to the GPU, the Fortran code was converted to standard C code. Then, CUDA programing language, which was tailor-made for NVIDIA, was added to convert the standard C code into CUDA C for data-parallel execution on GPU, as GPU-HADVPPM. It prepared the input data for GPU-HADVPPM by constructing random numbers and tested its offline performance on the GPU platform.*

*Lines 342-349: The validation and evaluation of porting the HADVPPM scheme from the CPU*

*to the GPU platform were conducted using offline and coupling performance experiments. First, we validated the result between different CAMx versions, and then the offline performance of the GPU-HADVPPM on a single GPU was tested by offline experiment. Finally, the coupling performance experiments illustrate its potential in three dimensions with varying chemical regimes. Sect.4.2 and Sect.4.4, the CAMx version of the HADVPPM scheme written by Fortran language, standard C, and CUDA C, is named F, C, and CUDA C, respectively.*

6. Line 354: Please clarify "30-min spatial resolution".

Response: Sorry for the confusion. Minute is the angular resolution. According to the conversion formula, 30-min is 0.5 degrees, and the corresponding range resolution is about 55.6 kilometers at mid-latitude. We have revised this part in line 361, which are as follows:

*30-min (about 55.6km at mid-latitude) spatial resolution*

Reference

Mielikainen, J., Huang, B., Huang, H.-L. A., and Goldberg, M. D.: GPU Implementation of Stony Brook University 5-Class Cloud Microphysics Scheme in the WRF, IEEE Journal of Selected Topics in Applied Earth Observations and Remote Sensing, 5, 625-633, 10.1109/jstars.2011.2175707, 2012.

NVIDIA: Floating Point and IEEE 754 Compliance for NVIDIA GPUs. Release 12.1, available at: https://docs.nvidia.com/cuda/floating-point/#differences-from-x86 (last access: 18 May 2023), 2023.

Wang, P., Jiang, J., Lin, P., Ding, M., Wei, J., Zhang, F., Zhao, L., Li, Y., Yu, Z., Zheng, W., Yu, Y., Chi, X., and Liu, H.: The GPU version of LASG/IAP Climate System Ocean Model version 3 (LICOM3) under the heterogeneous-compute interface for portability (HIP) framework and its large-scale application, Geosci. Model Dev., 14, 2781-2799, 10.5194/gmd-14-2781-2021, 2021.

Wang, Z., Wang, Y., Wang, X., Li, F., Zhou, C., Hu, H., and Jiang, J.: GPU-RRTMG_SW: Accelerating a Shortwave Radiative Transfer Scheme on GPU, IEEE Access, 9, 84231-84240, 10.1109/access.2021.3087507, 2016.

---

## Author Response (AR2)

**Response to Topical Editor**

Please submit the final version that has been revised in accordance with the two reviewers' comments. It is recommended to invite a native speaker to correct and polish the text.

**Dear Editor,**

Thanks a million for your precious time and your suggestion. In order to improve the English language of the manuscript, we called for the English language services from AJE. The certificate is shown as followed. More, the point-to-point responses to the reviewers' comments are listed below.

[Figure]

**Response to Reviewers' comments**

We are thankful to the two reviewers for their thoughtful and constructive comments that help us improve the manuscript substantially. We have revised the manuscript accordingly. Listed below is our point-to-point response in blue to each comment that was offered by the reviewers.

**Response to Reviewer #1**

General comments:

This paper presents implementation and optimization of air quality model CAMx using Cuda C targeting GPU clusters. Experiment results show that GPU-HADVPPM can achieve about 1000x acceleration, the series of optimization can acheive about dozens of times acceleration and the final

version of CAMx-GPU can achieve 4.5x speedup with 8 CPU cores and 8 GPU accelerators on V100 cluster. Here are some specific comments.

Response: We appreciate the editor for reviewing our manuscript and for the valuable suggestions, which we will address point by point in the following.

1. This paper needs to be more convincing about how some experiment results will be explained. Such as: the offline experiment achieves about 1000x acceleration which is far exceed the ratio of theoretical peak performance of GPU and CPU.

   Response: Thanks for the constructive comment. In the thread hierarchy, the thread is the most basic GPU unit for parallel computing. Threads can be organized into one-dimensional, two-dimensional, or three-dimensional blocks. In a three-dimensional block, each dimension of the three-dimensional block can contain a maximum number of threads of 1024, 1024 and 64. Similarly, blocks can be organized into one-dimensional, two-dimensional, or three-dimensional grids. In a three-dimensional grid, each dimension of the three-dimensional grid can contain a maximum number of blocks of $2^{31}$-1, 65535, and 65535. It is theoretically possible to distribute a large number of copies of kernel functions into tens of billions of threads for parallel computing without exceeding the GPU memory. In the offline performance test, we adopted a one-dimensional thread organization, with 1024 threads per block and several blocks based on the array size. For example, in the data size of $10^7$, the GPU can implement the parallel computation of 10 million threads, while the CPU can only use serial cyclic computation. Therefore, the offline experiments achieve up to 1100x acceleration, and the larger the data size, the more pronounced the advantage of GPU multithreading architecture and the higher the computing efficiency. In addition, the data transfer time between CPU and GPU is not considered in the offline performance experiments. Also, without I/O, the GPU-based SBU-YLIN scheme in the WRF model can achieve a **896x acceleration** compared to the Fortran implementation running on the CPU (Mielikainen et al., 2012b). We have revised this part in lines 476-487, which are as follows:

   Lines 476-487: *As described in Sect. 3.2, the thread is the most basic GPU unit for parallel*

*computing. Each dimension of the three-dimensional block can contain a maximum number of threads of 1024, 1024 and 64. Each dimension of the three-dimensional grid can contain a maximum number of blocks of $2^{31}$-1, 65535, and 65535. It is theoretically possible to distribute a large number of copies of kernel functions into tens of billions of threads for parallel computing without exceeding the GPU memory. In the offline performance experiments, the GPU achieved up to 10 million threads of parallel computing, while the CPU can only use serial cyclic computation. Therefore, GPU-HADVPPM achieves a maximum acceleration of approximately 1100x without I/O. In addition to this study, the GPU-based SBU-YLIN scheme in the WRF model can achieve a 896x acceleration compared to the Fortran implementation running on the CPU (Mielikainen et al., 2012b).*

2. It is suggested to supplement performance data of CAMx implemented by Fortran and C in section4.4.1.

   Response: Thanks for the constructive comment. We supplement the coupling performance experiment results for the C version of HADVPPM. On the K40m and V100 cluster, the elapsed time of the C version HADVPPM is 51.4 seconds and 45.2 seconds, respectively, 26.7% and 33.4% lower than that of the Fortran version HADVPPM. We have revised this part in lines 533-549, which are as follows:

   Lines 533-549: *In terms of the single module computational efficiency of HADVPPM and GPU-HADVPPM, we further tested the computational performance of the Fortran version of HADVPPM on the CPU, C version of HADVPPM on the CPU, and the CUDA C version of GPU-HADVPPM in CAMx-CUDA V1.4 (GPU-HADVPPM V1.4) on the GPU using system_clock functions in the Fortran language and cudaEvent_t in CUDA programming. The specific results are shown in Figure 10. On the K40m cluster, it takes 37.7 seconds and 51.4 seconds to launch the Intel Xeon E5-2682 v4 CPU to run the Fortran and C version HADVPPM, respectively, and the C version is 26.7% slower than the Fortran version. After the CUDA technology was used to convert the C code into CUDA C, the CUDA C version took 29.6 seconds to launch an NVIDIA Telsa K40m GPU to run GPU-HADVPPM V1.4, with a 1.3x and 1.7x acceleration. On the V100 cluster, the Fortran, C, and CUDA C versions are computationally*

*more efficient than those on the K40m cluster. It takes 30.1 seconds and 45.2 seconds to launch the Intel Xeon Platinum 8168 CPU to run the Fortran and C version HADVPPM, and 1.6 seconds to run the GPU-HADVPPM V1.4 using an NVIDIA V100 GPU. The computational efficiency of the CUDA C version is 18.8x and 28.3x higher than the Fortran and C versions, respectively.*

[Figure]

*Figure 10. The elapsed time of the Fortran version HADVPPM on the CPU, the C version HADVPPM on the CPU and the CUDA C version GPU-HADVPPM V1.4 on the GPU. The unit is in seconds (s).*

3. There are some typing errors, such as CAS-EMS should be CAS-ESM, CAMx-V1.0 in line 492 should be CAMx-CUDA V1.0.

Response: Sorry for this mistake. We have corrected typing errors in line 68, 88, and 518, which are as follows:

Line 68: *The GPU has proven successful in weather models such as the nonhydrostatic icosahedral model (NIM; Govett et al., 2017), global/regional assimilation and prediction system (GRAPES; Xiao et al., 2022), weather research and forecasting model (WRF; Huang et al., 2011; Huang et al., 2012; Mielikainen et al., 2012a; Mielikainen et al., 2012b; Mielikainen et al., 2013a; Mielikainen et al., 2013b; Price et al., 2014; Huang et al., 2015), ocean models such as the LASG/IAP climate system ocean model (LICOM; Jiang et al., 2019; Wang et al.,*

*2021a) and Princeton ocean model (POM; Xu et al., 2015) and earth system model of the Chinese Academy of Sciences (**CAS-ESM**; Wang et al., 2016; Wang et al., 2021b).*

Line 88: *In terms of climate system models, Wang et al. (2016) and Wang et al. (2021b) used CUDA Fortran and CUDA C to conduct heterogeneous porting of the RRTMG_SW and RRTMG_LW schemes of the atmospheric component model of the **CAS-ESM** earth system model and achieved a 38.88x and 77.78x acceleration, respectively.*

Line 518: *In CAMx-CUDA V1.2, the data transmission frequency between CPU-GPU within one time step is reduced to 1, the elapsed time on the two heterogeneous clusters is 1207 seconds and 548 seconds, respectively, and the speedup is 9.0x and 68.0x compared to **CAMx-CUDA V1.0**.*

Reference

Mielikainen, J., Huang, B., Huang, H.-L. A., and Goldberg, M. D.: GPU Implementation of Stony Brook University 5-Class Cloud Microphysics Scheme in the WRF, IEEE Journal of Selected Topics in Applied Earth Observations and Remote Sensing, 5, 625-633, 10.1109/jstars.2011.2175707, 2012.

**Response to Reviewer #2**

GPU accelerators have great potential to improve the efficiency of air quality modeling. The authors have explained the steps taken to convert a PPM solver of horizontal advection for air quality model CAMx into a new Compute Unified Device Architecture C (CUDA C) code and how this optimization approach has resulted in significant improvement in computational efficiency. This paper presents a promising approach for using GPU accelerators in air quality modeling and provides valuable insights into the optimization measures that can be taken to improve the overall computing performance of model. The authors' findings can potentially benefit researchers in air quality modeling and related fields. My comments listed below.

Response: We appreciate the editor for reviewing our manuscript and for the valuable suggestions, which we will address point by point in the following.

1.  The manuscript should be written more concisely, removing unnecessary material. For example, the information presented in Table 1 partly duplicates that presented in Figure 1b. Figure 9 doesn't provide too much important information and can be moved to Supplementary.

    Response: Thanks for the constructive comment. We have removed Table 1, and placed Figure 9 in the supplementary. The relevant expressions and contents in the paper have also been modified in lines 156-162 and 418-421, which are as follows:

    Lines 156-162: *The VTune tool detects each module's runtime and the most time-consuming functions on P1. As shown in Figure 1b, the top four time-consuming modules are chemistry, diffusion, horizontal advection and vertical advection in the CAMx model. In the above four modules, the top five most time-consuming programs are the ebirate, hadvppm, tridiag, diffus and ebisolv programs, and the total runtime of P1 is 325.1 seconds. Top1 and Top2's most time-consuming programs take 49.4 and 35.6 seconds, respectively.*

    Lines 418-421: *In terms of the time series, the regionally averaged time series of the three versions are almost consistent (as shown in Figure S2), and the maximum AEs for the above six species are 0.001 ppbv, 0.005 ppbv, 0.002 ppbv, 0.03 ppbv, 0.0001 ppbv and 0.0002 $\mu g \cdot m^{-3}$, respectively, between the Fortran and CUDA C versions.*

2.  The authors select the hadvppm program to implement the heterogeneous porting. It could benefit from providing some additional context about the motivation for selecting hadvppm program.

    Response: Thanks for the constructive comment. There are some reasons for selecting hadvppm for heterogeneous porting. First, the advection module is one of the air quality model's compulsory modules, and is mainly used to simulate the transport process of air pollutants, additionally it is also a hotspot module detected by the Intel VTune tool. The typical air quality models, CAMx, CMAQ and NAQPMS, include advection modules and use the

exact PPM advection solver. The heterogeneous version developed in this study can be directly applied to the above models. Furthermore, the weather model (e.g., WRF) also contains an advection module, so this study's heterogeneous porting method and experience can be used for reference. We have revised this part in lines 163-173, which are as follows:

Lines 163-173: *By consideration, the hadvppm program was selected to conduct heterogeneous porting for several reasons. First, the advection module is one of the air quality model's compulsory modules, and is mainly used to simulate the transport process of air pollutants, additionally it is also a hotspot module detected by the Intel VTune tool. The typical air quality models, CAMx, CMAQ and NAQPMS, include advection modules and use the exact PPM advection solver. The heterogeneous version developed in this study can be directly applied to the above models. Furthermore, the weather model (e.g., WRF) also contains an advection module, so this study's heterogeneous porting method and experience can be used for reference. Therefore, a GPU acceleration version of the HADVPPM scheme, namely, GPU-HADVPPM, is built to improve the CAMx performance.*

3. In Figure 5-7, large absolute errors mainly occur over Zhangjiakou City for $SO_2$, $NO_2$ CO and $PSO_4$ while simulated $O_3$ and $H_2O_2$ show large errors over Baoding city. Please explain how do porting process cause such patterns.

Response: Thanks for the constructive comment. During the process of heterogeneous porting, the CUDA technology was used to convert the standard C code into CUDA C to make the hadvppm program computable on the GPU. However, due to the slight difference in data operation and accuracy between CPU and GPU(NVIDIA,2023), the concentration variable of hadvppm program appears to have minimal negative values (about $-10^{-4} \sim -10^{-9}$) when integrating on GPU. In order to allow the program to continue running, we forcibly replace these negative values with $10^{-9}$. It is because these negative values are replaced by positive values that the simulation results are biased. Furthermore, we adopted the evaluation method of Wang et al. (2021) to compute the ratio between the root-mean-square-error and standard deviation of six species, such as $SO_2$, $O_3$, $NO_2$, CO, $H_2O_2$, and $PSO_4$. The results were 0.004%, 0.003%, 0.003%, 0.006%, 0.015%, and 0.004%, respectively, which were far less than the 0.2%

mentioned by Wang et al. (2021), so the absolute errors of simulation results in this study are within the acceptable range. We have revised this part in lines 376-384, which are as follows:

Lines 376-384: *During the porting process, the primary error comes from converting standard C to CUDA C, and the main reason is related to the hardware difference between the CPU and GPU. Due to the slight difference in data operation and accuracy between the CPU and GPU (NVIDIA,2023), the concentration variable of the hadvppm program appears to have minimal negative values (approximately $-10^{-9} \sim -10^{-4}$) when integrated on the GPU. To allow the program to continue running, we forcibly replace these negative values with $10^{-9}$. It is because these negative values are replaced by positive values that the simulation results are biased.*

4. I'd suggest the authors discussing in the Conclusions and Discussion section any limitations of the current approach, which may warrant future refinement.

Response: Thanks for the constructive comment. There are some limitations of the current approach.

1) We currently implement thread and block coindexing to compute horizontal grid points in parallel. Given the CAMx Model 3-dimensional grid computing characteristics, in the future, 3-dimensional thread and block coindexing will be considered to compute 3-dimensional grid points in parallel.

2) The communication bandwidth of data transfer is one of the main issues restricting the computing performance of the CUDA C codes on the GPUs. This restriction holds true not only for GPU-HADVPPM but also for the WRF module (Mielikainen et al., 2012b; Mielikainen et al., 2013b; Huang et al., 2013). In this study, the data transmission efficiency between the CPU and GPU is improved only by reducing the communication frequency. In the future, more technologies, such as pinned memory (Wang et al., 2016), will be considered to resolve the communication bottleneck between the CPUs and GPUs.

3) To further improve the overall computational efficiency of the CAMx model, the heterogeneous porting scheme proposed in this study will be considered to conduct the heterogeneous porting of other CAMx modules in the future.

We have added this part in lines 600-615, which are as follows:

Lines 600-615: *However, the current approach has some limitations, which are as follows:*

*4)    We currently implement thread and block coindexing to compute horizontal grid points in parallel. Given the CAMx Model 3-dimensional grid computing characteristics, in the future, 3-dimensional thread and block coindexing will be considered to compute 3-dimensional grid points in parallel.*

*5)    The communication bandwidth of data transfer is one of the main issues restricting the computing performance of the CUDA C codes on the GPUs. This restriction holds true not only for GPU-HADVPPM but also for the WRF module (Mielikainen et al., 2012b; Mielikainen et al., 2013b; Huang et al., 2013). In this study, the data transmission efficiency between the CPU and GPU is improved only by reducing the communication frequency. In the future, more technologies, such as pinned memory (Wang et al., 2016), will be considered to resolve the communication bottleneck between the CPUs and GPUs.*

*6)    To further improve the overall computational efficiency of the CAMx model, the heterogeneous porting scheme proposed in this study will be considered to conduct the heterogeneous porting of other CAMx modules in the future.*

5.    Line 173-177, 337-342: Please unify the tense of the sentence.

Response: Sorry for these mistakes. We have revised this part in lines 180-187 and 342-349, which are as follows:

Lines 180-187: *The CAMx-CUDA heterogeneous scheme is shown in Figure 2. The second time-consuming hadvppm program in the CAMx model was selected to implement heterogeneous porting. To map the hadvppm program to the GPU, the Fortran code was converted to standard C code. Then, the CUDA programming language, which was tailor-made for NVIDIA, was added to convert the standard C code into CUDA C for data-parallel execution on the GPU, as GPU-HADVPPM. It prepared the input data for GPU-HADVPPM by constructing random numbers, and tested its offline performance on the GPU platform.*

Lines 343-350: *The validation and evaluation of porting the HADVPPM scheme from the CPU to the GPU platform were conducted using offline and coupling performance experiments. First, we validated the results between the different CAMx versions, and then the offline performance of the GPU-HADVPPM on a single GPU was tested by offline experiments. Finally, coupling performance experiments illustrate its potential in three dimensions with varying chemical regimes. In Sect.4.2 and Sect.4.4, the CAMx versions of the HADVPPM scheme written in Fortran, standard C and CUDA C are named F, C and CUDA C, respectively.*

6. Line 354: Please clarify "30-min spatial resolution".

   Response: Sorry for the confusion. Minute is the angular resolution. According to the conversion formula, 30-min is 0.5 degrees, and the corresponding range resolution is approximately 55.6 kilometers at mid-latitude. We have revised this part in line 362, which are as follows:

   Line 362: *30-min (approximately 55.6 km at mid-latitude) spatial resolution*

Reference

NVIDIA: Floating Point and IEEE 754 Compliance for NVIDIA GPUs. Release 12.1, available at: https://docs.nvidia.com/cuda/floating-point/#differences-from-x86 (last access: 18 May 2023), 2023.

Wang, P., Jiang, J., Lin, P., Ding, M., Wei, J., Zhang, F., Zhao, L., Li, Y., Yu, Z., Zheng, W., Yu, Y., Chi, X., and Liu, H.: The GPU version of LASG/IAP Climate System Ocean Model version 3 (LICOM3) under the heterogeneous-compute interface for portability (HIP) framework and its large-scale application, Geosci. Model Dev., 14, 2781-2799, 10.5194/gmd-14-2781-2021, 2021.

Wang, Z., Wang, Y., Wang, X., Li, F., Zhou, C., Hu, H., and Jiang, J.: GPU-RRTMG_SW: Accelerating a Shortwave Radiative Transfer Scheme on GPU, IEEE Access, 9, 84231-84240, 10.1109/access.2021.3087507, 2016.